# Inflammatory diseases and risk of lung cancer among individuals who have never smoked

Monica E. D'Arcy [1,2,3] ✉, Ruth M. Pfeiffer[1] ✉, Marie C. Bradley[1], Phuc H. Hoang [1], Thi-Van-Trinh Tran[1], John P. McElderry[1], Mengying Li[1], Michael Kebede[4], Curt T. DellaValle [5], Sara Rivas[1], Youjin Wang[1], Shahinaz M. Gadalla [1] & Maria Teresa Landi [1] ✉

Lung cancer in never-smokers (LCINS) is a leading cause of cancer death globally, but no screening programs for LCINS exist. To identify medical conditions that could serve as markers of LCINS risk, we conducted a nested case-control study within the United Kingdom's Clinical Practice Research Datalink (CPRD-GOLD), consisting of 1581 LCINS cases and 14,318 never-smoking controls. Conditions significantly associated with LCINS 1-10 years before the index date were validated in an independent dataset, CPRD-Aurum (2188 LCINS cases, 19,597 never-smoking controls). These conditions include Chronic Obstructive Pulmonary Disease/Emphysema (COPD); gastro-esophageal reflux disease (GERD); bronchitis and tracheitis; diabetes mellitus type 1; and gastritis and non-infective gastroenteritis and colitis. Adjusting for medication use only slightly attenuated these associations. Overall, inflammatory diseases appear to be important in LCINS pathogenesis although further studies need to confirm these associations. Conditions such as GERD or COPD could be considered as part of eligibility criteria for future LCINS screening programs.

Lung cancer (LC) is the leading cause of cancer-related mortality globally and in the United States (US). Although a majority of LC cases are attributable to smoking, 15–25% of all lung cancers in the US occur in persons who have never smoked[1]. Moreover, LC in individuals who have never smoked (LCINS) is the 8th leading cause of cancer-related mortality in the US (5th globally)[2]. Because persons who have never smoked (PWHNS) are ineligible for LC screening, they are commonly diagnosed at advanced stages, when 5-year survival is only 26%[3,4].

There are notable differences between LC arising in smokers and LC in PWHNS. LCINS is twice as common in women—particularly Asian and Hispanic women—than in men. Further, LC in younger persons is much more likely to occur in PWHNS[2,5,6]. LCINS appears to be histologically and molecularly distinct from LC in smokers, with a greater proportion of LCINS cases being of adenocarcinoma histology and harboring targetable driver mutations[2,5–7]. Factors associated with LCINS appear to selectively target distal airways, unlike smoking-related carcinogens that target both proximal and distal lung compartments[6].

There are several known or suspected risk factors for LCINS, including exposure to second-hand tobacco smoke, indoor and out-door air pollution, radon, asbestos, silica, and heavy metals, and inherited genetic susceptibility[5,6,8–11]. Previous lung diseases likely also increase LCINS risk. A large, pooled analysis found significantly elevated rates of LCINS among individuals diagnosed with pneumonia

[1]Division of Cancer Epidemiology and Genetics, National Cancer Institute, Rockville, MD, USA. [2]Rutgers Center for Pharmacoepidemiology and Treatment Sciences, New Brunswick, NJ, USA. [3]Rutgers Cancer Institute, New Brunswick, NJ, USA. [4]Department of Epidemiology, Gillings School of Global Public Health, University of North Carolina, Chapel Hill, NC, USA. [5]Division of Cancer Control and Population Sciences, National Cancer Institute, Rockville, MD, USA. ✉e-mail: monica.darcy@rutgers.edu; pfeiffer@mail.nih.gov; landim@nih.gov

and tuberculosis 5-10 years prior to the lung cancer diagnosis[12]. These risk factors only explain a small proportion of LCINS in the US and Western Europe[13].

Because the majority of LCINS cannot be explained by established risk factors, alternative approaches are needed to gain etiologic insight. Electronic Medical Records (EMRs) contain extensive health-related data on medical diagnoses, laboratory tests, and prescription drug use. These data can be leveraged to screen multiple medical conditions to generate hypotheses for further follow-up[14]. The aim of this study was to use the United Kingdom's (UK) Clinical Practice Research Datalink (CPRD) to agnostically identify and independently validate medical conditions associated with the risk of LCINS. This study is part of the Sherlock-Lung project on LCINS[15,16]. In this work, we show that individuals with LCINS are more likely to have been diagnosed with inflammatory diseases. In addition to providing etiologic insights, these findings may help lay a foundation for future risk stratification and LCINS screening programs.

## Results

### Discovery stage

We identified 1581 LCINS cases and 14,318 never-smoking controls who had been registered for a median of 15.0 years (interquartile range [IQR] 6.3–28.3) prior to the index date (Table 1). Differences between the feasibility study ("Methods") and the actual numbers of included cases arose because the final study population included an additional year (2019) and we imposed restrictions (e.g., age at diagnosis 30–89). By design, cases and controls were similar with respect to age, sex, registration year, and region, although the proportion of female controls was slightly higher than female cases (58.2% versus 56.4%).

In the 1–10 years prior to the index date (the date of diagnosis for cases and the date of selection for controls), infections and inflammation (II) (adjusted odds ratio [aOR] = 1.26, 95% confidence interval [CI] 1.12–1.42) and anemia (aOR = 1.44, 95%CI 1.13–1.83) were associated with LCINS in conditional logistic regression analyses (Table 2). Results for all categories and subcategories are listed in Supplementary Table S1. Dementia and Alzheimer's disease were inversely associated with LCINS risk (aOR = 0.55, 95%CI 0.36–0.84). However, these associations could be confounded by functional status (i.e., individuals who are frail and/or have short life expectancy are unlikely to be evaluated and/or treated for cancer) and we thus elected not to validate them. Although the cardiovascular disease category was not associated with LCINS at a false discovery rate (FDR) $p$-value < 0.05, a few diseases within that category were strongly associated with LCINS risk, including: peripheral vascular disease (aOR = 2.93, 95%CI 1.74–4.96) and myocardial infarction (aOR = 1.51, 95%CI 1.22–1.88). Because the category was not associated with LCINS overall, these conditions were not considered for validation. Thrombosis, kidney disease, eye conditions, and osteoporosis were significantly associated (aOR$_{\text{thrombosis 10-32 years prior}}$ = 2.28, 95%CI 1.47-3.52, aOR$_{\text{kidney disease 10-32 years prior}}$ = 2.67, 95%CI 1.22–5.87, aOR$_{\text{eye conditions 10-32 years prior}}$ = 1.39, 95%CI 1.00–1.92, aOR$_{\text{osteoporosis 10-32 years prior}}$ = 1.57, 95%CI 1.01–2.46) with LCINS in the 10–32 years prior to the index date despite not being significantly associated with LCINS in the 1–10 years prior to the index date (aOR$_{\text{thrombosis 1-10 years prior}}$ = 1.20, 95%CI 0.87-1.64, aOR$_{\text{kidney disease 1-10 years prior}}$ = 1.07, 95%CI 0.87–1.33, aOR$_{\text{eye conditions 1-10 years prior}}$ = 1.39, 95%CI 1.00–1.92, aOR$_{\text{osteoporosis 10-32 years prior}}$ = 1.57, 95%CI 1.01–2.46). No other primary categories were associated with LCINS in the 10–32 years prior to the index date.

In hierarchical logistic models, the group mean estimate for the Infections and Inflammation (II) category was similar to that from the conditional logistic regression analysis (aOR = 1.22, 95%CI 1.08-1.38). Of the 51 II sub-categories, 12 were associated with LCINS in the 1–10 years prior to the index date (Table 3 and Fig. 1). In hierarchical models,

**Table 1 | Characteristics of lung cancer cases who have never smoked diagnosed between 1998-2019 and matched controls who have never smoked in CPRD-GOLD and CPRD-Aurum**

| Characteristic | Discovery[a] data (CPRD-GOLD) | | Validation[a] data (CPRD-Aurum) | |
| --- | --- | --- | --- | --- |
| | Cases (n = 1581) | Controls (n = 14,318) | Cases (n = 2188) | Controls (n = 19,597) |
| Age (years) at selection/diagnosis, n (%) | | | | |
| 30–39 | 18 (1.1) | 198 (1.4) | 38 (1.7) | 362 (1.8) |
| 40–49 | 77 (4.9) | 689 (4.8) | 92 (4.2) | 917 (4.7) |
| 50–59 | 168 (10.6) | 1581 (11.0) | 249 (11.4) | 2359 (12.0) |
| 60–69 | 356 (22.5) | 3240 (22.6) | 445 (20.3) | 4189 (21.4) |
| 70–79 | 522 (33.0) | 4560 (31.8) | 681 (31.1) | 6113 (31.2) |
| 80–89 | 440 (27.9) | 4050 (28.3) | 683 (31.2) | 5657 (28.9) |
| Female, n (%) | 891 (56.4) | 8338 (58.2) | 1265 (57.8) | 11471 (58.5) |
| Year of practice registration, n (%) | | | | |
| 1980 and earlier | 409 (26.0) | 3588 (25.1) | 551 (25.2) | 4612 (23.5) |
| 1981–1990 | 325 (20.6) | 2893 (20.2) | 479 (21.9) | 4351 (22.2) |
| 1991–2000 | 430 (27.2) | 3793 (26.5) | 618 (28.2) | 5663 (28.9) |
| 2001–2010 | 335 (21.2) | 3246 (22.7) | 403 (18.4) | 3697 (18.9) |
| 2011–2018 | 82 (5.2) | 798 (5.6) | 137 (6.3) | 1274 (6.5) |
| Index year, n (%) | | | | |
| 1989–1994 | 43 (2.7) | 343 (2.4) | 9 (0.4) | 74 (0.4) |
| 1995–2000 | 173 (10.9) | 1455 (10.2) | 256 (11.7) | 2322 (11.8) |
| 2001–2006 | 397 (25.1) | 3221 (22.5) | 523 (23.9) | 4645 (23.7) |
| 2007–2012 | 534 (33.8) | 5024 (35.1) | 671 (30.7) | 5985 (30.5) |
| 2013–2019 | 434 (27.5) | 4275 (29.9) | 729 (33.3) | 6571 (33.5) |
| Duration of registration in practice, year (IQR) | 15.1 (6.3, 28.5) | 14.9 (6.3, 28.3) | 16.2 (7.4, 28.3) | 15.6 (7.32,27.4) |
| Body Mass Index (kg/m2), n (%) | | | | |
| Under-weight (< 18.5) | 19 (1.2) | 196 (1.4) | 41 (1.9) | 269 (1.4) |
| Normal weight (18.5–24.9) | 459 (29.0) | 3937 (27.5) | 671 (30.7) | 5774 (29.5) |
| Overweight (25.0–29.9) | 467 (29.5) | 4246 (29.7) | 722 (33.0) | 5857 (29.9) |
| Obese (≥ 30.0) | 248 (15.7) | 2294 (16.0) | 331 (15.1) | 3250 (16.6) |
| Missing | 388 (24.5) | 3645 (25.5) | 423 (19.3) | 4447 (22.7) |
| Alcohol Consumption, n (%) | | | | |
| Current | 824 (52.1) | 7103 (49.6) | 1574 (71.9) | 13204 (67.4) |
| Former | 95 (6.0) | 804 (5.6) | 34 (1.6) | 394 (2.0) |
| Never | 267 (16.9) | 2252 (15.7) | 134 (6.1) | 1021 (5.2) |
| Missing | 395 (25.0) | 4159 (29.0) | 446 (20.4) | 4978 (25.4) |
| Region, n (%) | | | | |
| East Midlands | 29 (1.8) | 276 (1.9) | 60 (2.7) | 543 (2.8) |
| East of England | 73 (4.6) | 679 (4.7) | 133 (6.1) | 1226 (6.3) |
| London | 177 (11.2) | 1563 (10.9) | 271 (12.4) | 2325 (11.9) |
| North East | 9 (22.5) | 56 (0.4) | 105 (4.8) | 1027 (5.2) |
| North West | 138 (33.0) | 1245 (8.7) | 395 (18.1) | 3338 (17.0) |
| Northern Ireland | 45 (2.8) | 458 (3.2) | – | – |
| Scotland | 355 (22.5) | 3153 (22.0) | – | – |
| South Central | 111 (7.0) | 999 (7.0) | 221 (10.1) | 1987 (10.1) |
| South East Coast | 144 (9.1) | 1334 (9.3) | 175 (8.0) | 1494 (7.6) |
| South West | 100 (6.3) | 925 (6.5) | 330 (15.1) | 3085 (15.7) |
| Wales | 256 (16.2) | 2388 (16.7) | – | – |
| West Midlands | 112 (7.1) | 976 (6.8) | 398 (18.2) | 3589 (18.3) |
| Yorkshire and the Humber | 32 (2.0) | 266 (1.9) | 100 (4.6) | 983 (5.0) |

*IQR* interquartile range.
[a]Small differences in matching factors were attributable to the identification of individuals reporting a history of smoking after the selection date. We excluded these persons from the study.

**Table 2 | Adjusted odds ratios (aORs) and 95% confidence intervals (CIs) estimated from conditional logistic regression models for the association between primary disease categories and LCINS sorted by the false discovery rate for the 1-10 year associations in the discovery dataset (CPRD-GOLD)**

| Disease/disease category[c] | Discovery[a] (1-10 YEARS PRIOR TO SELECTION) | | | | | Discovery[b] (10-32 YEARS PRIOR TO SELECTION) | | | |
|---|---|---|---|---|---|---|---|---|---|
| | No. (%) in cases | No. (%) in controls | OR (95% CI) | p | FDR | No. (%) in cases | No. (%) in controls | OR (95% CI) | p |
| **Infections and Inflammation** | 1022 (64.6) | 8703 (60.8) | **1.26 (1.12, 1.42)** | **0.0001** | **0.003** | 405 (40.2) | 3701 (40.7) | 1.06 (0.90, 1.25) | 0.465 |
| **Anemia** | 87 (5.5) | 559 (3.9) | **1.44 (1.13, 1.83)** | **0.0029** | **0.035** | 24 (2.4) | 190 (2.1) | 1.14 (0.74,1.78) | 0.552 |
| **Dementia/Alzheimer's Disease** | 26 (1.6) | 404 (2.8) | **0.55 (0.36, 0.84)** | **0.0055** | **0.044** | NR | 14 (0.2) | 0.70 (0.09, 5.40) | 0.730 |
| Hernia | 134 (8.5) | 961 (6.7) | 1.29 (1.06, 1.56) | 0.011 | 0.064 | 50 (5.0) | 431 (4.7) | 1.09 (0.80, 1.48) | 0.595 |
| Cardiovascular Disease | 890 (56.3) | 7717 (53.9) | 1.14 (1.01, 1.28) | 0.030 | 0.143 | 347 (34.4) | 3212 (35.3) | 1.02 (0.86,1.22) | 0.792 |
| Thyroid Disease | 40 (2.5) | 278 (1.9) | 1.41 (1.00, 1.99) | 0.049 | 0.196 | 11 (1.1) | 79 (0.9) | 1.24 (0.64, 2.39) | 0.520 |
| Polyps, Cysts, Fibrosis | 177 (11.2) | 1424 (9.9) | 1.17 (0.99, 1.39) | 0.070 | 0.239 | 86 (8.5) | 650 (7.1) | 1.25 (0.98, 1.61) | 0.071 |
| Female Conditions | 61 (3.9) | 487 (3.4) | 1.22 (0.92, 1.62) | 0.177 | 0.526 | 35 (3.5) | 323 (3.6) | 0.98 (0.67, 1.44) | 0.913 |
| In situ cancers | 17 (1.1) | 117 (0.8) | 1.41 (0.84, 2.36) | 0.197 | 0.526 | NR | 43 (0.5) | 0.63 (0.19, 2.05) | 0.443 |
| **Thrombosis** | 46 (2.9) | 371 (2.6) | 1.20 (0.87, 1.64) | 0.261 | 0.557 | 27 (2.7) | 119 (1.3) | **2.28 (1.47, 3.52)** | **0.0002** |
| Allergies | 337 (21.3) | 2897 (20.2) | 1.08 (0.94, 1.23) | 0.268 | 0.557 | 135 (13.4) | 1216 (13.4) | 1.06 (0.87, 1.30) | 0.551 |
| Osteoporosis | 92 (5.8) | 784 (5.5) | 1.13 (0.90, 1.43) | 0.292 | 0.557 | 25 (2.5) | 160 (1.8) | 1.57 (1.01, 2.46) | 0.047 |
| Psychiatric | 140 (8.9) | 1165 (8.1) | 1.11 (0.91, 1.34) | 0.301 | 0.557 | 67 (6.6) | 599 (6.6) | 1.08 (0.83, 1.42) | 0.557 |
| Neurological Disorders | 161 (10.2) | 1374 (9.6) | 1.09 (0.91, 1.29) | 0.363 | 0.617 | 58 (5.8) | 453 (5.0) | 1.23 (0.92, 1.65) | 0.162 |
| Diabetes Mellitus Type 2 | 145 (9.2) | 1237 (8.6) | 1.09 (0.90, 1.31) | 0.386 | 0.617 | 35 (3.5) | 284 (3.1) | 1.12 (0.78, 1.61) | 0.542 |
| Keratosis | 107 (6.8) | 935 (6.5) | 1.08 (0.87, 1.34) | 0.487 | 0.680 | 25 (2.5) | 217 (2.4) | 1.09 (0.71, 1.68) | 0.696 |
| Kidney Disease | 123 (7.8) | 1094 (7.6) | 1.07 (0.87, 1.33) | 0.518 | 0.680 | 11 (1.1) | 55 (0.6) | 2.67 (1.22,5.87) | 0.014 |
| Stroke Related | 81 (5.1) | 691 (4.8) | 1.08 (0.85, 1.38) | 0.531 | 0.680 | 13 (1.3) | 146 (1.6) | 0.83 (0.46, 1.48) | 0.526 |
| Conditions affecting males | 63 (4.0) | 600 (4.2) | 0.92 (0.70, 1.22) | 0.570 | 0.680 | 11 (1.1) | 161 (1.8) | 0.63 (0.33, 1.18) | 0.150 |
| Irritable Bowel Syndrome | 31 (2.0) | 321 (2.2) | 0.90 (0.62, 1.32) | 0.603 | 0.680 | 32 (3.2) | 208 (2.3) | 1.48 (1.00, 2.18) | 0.047 |
| Varicose Veins | 49 (3.1) | 405 (2.8) | 1.08 (0.80, 1.47) | 0.615 | 0.680 | 30 (3.0) | 250 (2.7) | 1.13 (0.77, 1.67) | 0.535 |
| Eye Conditions | 191 (12.1) | 1691 (11.8) | 1.04 (0.88, 1.24) | 0.623 | 0.680 | 48 (4.8) | 332 (3.7) | 1.39 (1.00, 1.92) | 0.048 |
| Gallbladder | 36 (2.3) | 327 (2.3) | 1.05 (0.74, 1.49) | 0.791 | 0.825 | 18 (1.8) | 178 (2.0) | 0.97 (0.59, 1.59) | 0.891 |
| Osteoarthritis | 252 (15.9) | 2285 (16.0) | 1.01 (0.87, 1.17) | 0.926 | 0.926 | 97 (9.6) | 961 (10.6) | 0.91 (0.72, 1.15) | 0.422 |

aOR adjusted odds ratio, FDR false discovery rate, NR: Not reportable because of CPRD cell size (<5) reporting requirements.

Bolded rows indicate that the result is significant in the 1–10 years prior to selection (false discovery rate for multiple testing correction <0.05).

aOR: Conditional logistic regression models [Controls were individually matched (5–10) to cases on year of birth (+/–2 years); sex; general practice or region (general practice first, then region if we could not identify a control within the same practice); and year of practice registration (+/–2 year)] odds ratios are adjusted for age (linear term).

aIncludes 1581 never-smoking lung cancer cases and 14,318 never-smoking controls with at least 1 year of registration within a CPRD-GOLD primary care clinic.

bIncludes 1008 cases and 9093 controls with at least 10 years of registration within a CPRD-GOLD primary care clinic.

cRead codes that comprise each disease or disease category grouping are documented in supplemental material DiscoveryGoldCodes.xlsx. See Supplementary Table S1 for the hierarchy and listing of tab names associated with each disease/disease category. Multiple testing corrections were applied for conditions diagnosed 1–10 years prior to lung cancer in non-smokers (LCINS). No corrections were conducted for conditions diagnosed 10–32 years prior to LCINS, as these were secondary analyses.

**Table 3 | Statistically significant associations (adjusted odds ratios, aORs, and 95% confidence intervals, CIs, estimated from hierarchical logistic regression models) between disease subcategories in the Infection and Inflammation category and LCINS in the discovery dataset**

| Medical condition | Discovery (1–10 YEARS PRIOR TO SELECTION)[a] | | | | Discovery (10–32 YEARS PRIOR TO SELECTION)[b] | | | |
|---|---|---|---|---|---|---|---|---|
| | No. (%) in cases | No. (%) in controls | OR (95% CI) | p | No. (%) in cases | No. (%) in controls | OR (95% CI) | p |
| **Infections and Inflammation** (fixed effect) | 1022 (64.6) | 8703 (60.8) | 1.22 (1.08,1.38) | 0.002 | 405 (40.2) | 3701 (40.7) | 0.96 (0.81,1.13) | 0.606 |
| Tuberculosis | NR | 6 (0.0) | 3.69 (1.01,13.38) | 0.047 | NR | 18 (0.2) | 0.96 (0.81,1.13) | 0.606 |
| Influenza | 34 (2.2) | 198 (1.4) | 1.52 (1.09,2.10) | 0.013 | 24 (2.4) | 175 (1.9) | 0.96 (0.81,1.13) | 0.606 |
| Upper respiratory infections[c] | 325 (20.6) | 2599 (18.2) | 1.25 (1.09,1.43) | 0.002 | 69 (6.8) | 505 (5.6) | 1.05 (0.86,1.28) | 0.626 |
| **Autoimmune conditions**[d] | 222 (14.0) | 1674 (11.7) | 1.28 (1.10,1.49) | 0.002 | 60 (6.0) | 626 (6.9) | 0.89 (0.68,1.18) | 0.432 |
| Psoriasis | 30 (1.9) | 205 (1.4) | 1.29 (1.02,1.62) | 0.031 | 8 (0.8) | 95 (1.0) | 0.96 (0.81,1.13) | 0.606 |
| DMT1 | 14 (0.9) | 69 (0.5) | 1.74 (1.05,2.89) | 0.033 | NR | 30 (0.3) | 0.62 (0.24,1.63) | 0.333 |
| Lupus | 5 (0.3) | 11 (0.1) | 3.78 (1.36,10.54) | 0.011 | NR | 6 (0.1) | 3.30 (0.89,12.31) | 0.075 |
| Other AI not classified | 35 (2.2) | 190 (1.3) | 1.77 (1.24,2.53) | 0.002 | 13 (1.3) | 73 (0.8) | 1.32 (0.79,2.21) | 0.294 |
| Rosacea | 33 (2.1) | 210 (1.5) | 1.52 (1.09,2.12) | 0.013 | 6 (0.6) | 90 (1.0) | 0.95 (0.77,1.16) | 0.602 |
| **Gastrointestinal disease**[d] | 360 (22.8) | 2821 (19.7) | 1.25 (1.10,1.42) | 0.001 | 132 (13.1) | 1157 (12.7) | 1.08 (0.88,1.32) | 0.448 |
| Ulcer[e] | 20 (1.3) | 116 (0.8) | 1.27 (1.01,1.61) | 0.044 | 7 (0.7) | 71 (0.8) | 0.96 (0.81,1.13) | 0.606 |
| GERD | 94 (5.9) | 663 (4.6) | 1.41 (1.13,1.75) | 0.002 | 23 (2.3) | 193 (2.1) | 0.96 (0.81,1.13) | 0.606 |
| Gastritis/NIGC | 50 (3.2) | 284 (2.0) | 1.79 (1.31,2.45) | 0.000 | 21 (2.1) | 109 (1.2) | 1.53 (0.96,2.43) | 0.071 |
| COPD/Emphysema | 56 (3.5) | 176 (1.2) | 3.43 (2.47,4.76) | <.0001 | 11 (1.1) | 24 (0.3) | 3.28 (1.59,6.80) | 0.001 |

*aOR* adjusted odds ratio, *CI* confidence interval, *AI* autoimmune, *GERD* Gastroesophageal reflux disease, *DMT1* Diabetes Mellitus Type 1, *COPD* Chronic Obstructive Pulmonary Disorder, *NIGC* Non-infective gastroenteritis and colitis, *NR*: Not reportable because of CPRD cell size (<5) reporting requirements.

aOR: Hierarchical regression model odds ratios are adjusted for age (linear term), sex (categorical), region (categorical), registration year (continuous linear term).

[a]Study population includes 1581 never-smoking lung cancer cases and 14,318 never-smoking controls with at least 1 year of registration within a GOLD primary care clinic.

[b]Study population includes 1008 never-smoking lung cancer cases and 9093 controls with at least 10 years of registration within a GOLD primary care clinic.

[c]Consists of terms indicative of bronchitis, tracheitis, or upper respiratory infections.

[d]Estimates are from conditional logistic regression and are not included in hierarchical regression. See Supplementary Table S2.

[e]Consists of terms indicative of peptic, gastric, or duodenal ulcer.

Multiple testing correction is not needed because the ORs of the individual conditions are mutually adjusted for all other conditions within the primary category under the hierarchical model.

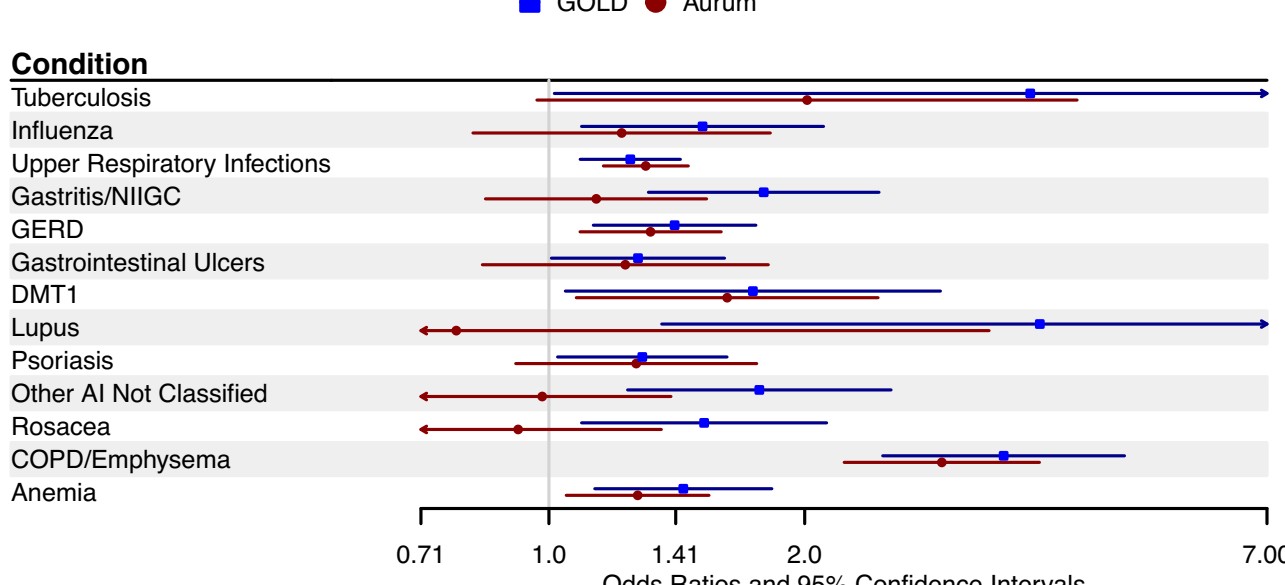

**Fig. 1 | Forest plots showing the associations between the medical conditions identified in the 1-10 years prior to selection and lung cancer in individuals who have never smoked.** Data are presented as adjusted odds ratios with 95% confidence intervals from conditional logistic regression models. Estimates in blue are the associations from the discovery (CPRD-GOLD) dataset (1581 LCINS cases and 14,318 never smoking controls), and those in red are the associations in the validation dataset (CPRD-Aurum) (2188 LCINS cases and 19,597 never smoking controls). Source data are provided as a Source Data File.

statistically significant associations were observed for COPD (aOR$_{1-10 \text{ years prior}}$ = 3.43, 95%CI 2.47–4.76); tuberculosis (aOR$_{1-10\text{years prior}}$ = 3.69, 95%CI 1.01–13.38); lupus (aOR$_{1-10 \text{ years prior}}$ = 3.78, 95%CI 1.36–10.54); gastrointestinal ulcers (aOR$_{1-10\text{years prior}}$ = 1.27, 95%CI 1.01–1.61); gastro-

esophageal reflux (GERD, aOR$_{1-10 \text{ years prior}}$ = 1.41, 95%CI 1.13–1.75); gastritis and non-infective gastroenteritis and colitis (gastritis/NIGC, aOR$_{1-10 \text{ years prior}}$ = 1.79, 95%CI 1.31–2.45); influenza (aOR$_{1-10 \text{ years prior}}$ = 1.52, 95%CI 1.09–2.10); upper respiratory infections, which largely consisted of

**Table 4 | Adjusted odds ratios and 95% confidence intervals from conditional logistic regression models for the associations between medical conditions and LCINS in the validation (CPRD-Aurum) dataset**

| | Validation (1-10 YEARS PRIOR TO SELECTION)[a] | | | | Validation (10-32 YEARS PRIOR TO SELECTION)[b] | | | |
|---|---|---|---|---|---|---|---|---|
| | No. (%) in cases | No. (%) in controls | OR (95% CI) | p | No. (%) in cases | No. (%) in controls | OR (95% CI) | p |
| **Infections and Inflammation** | | | | | | | | |
| **Tuberculosis** | **9 (0.4)** | **41 (0.2)** | **2.01 (0.97,4.18)** | **0.06** | 4 (0.3) | 17 (0.1) | 2.44 (0.81,7.35) | 0.11 |
| Influenza | 28 (1.3) | 207 (1.1) | 1.22 (0.81,1.82) | 0.34 | 25 (1.7) | 229 (1.8) | 0.92 (0.60,1.41) | 0.70 |
| **Upper Respiratory Infections[c]** | **444 (20.3)** | **3263 (16.7)** | **1.30 (1.16,1.46)** | **8.2E-06** | **200 (13.7)** | **1543 (11.9)** | **1.20 (1.01,1.42)** | **0.04** |
| **Gastritis/NIGC** | 50 (2.3) | 392 (2.0) | 1.14 (0.84,1.53) | 0.40 | **27 (1.9)** | **148 (1.1)** | **1.64 (1.08,2.49)** | **0.02** |
| **GERD** | 132 (6.0) | 917 (4.7) | 1.32 (1.09,1.60) | 4.8E-03 | 37 (2.5) | 300 (2.3) | 1.11 (0.78,1.58) | 0.56 |
| Gastrointestinal Ulcers[d] | 30 (1.4) | 209 (1.1) | 1.23 (0.84,1.81) | 0.29 | 16 (1.1) | 130 (1.0) | 1.05 (0.62,1.78) | 0.86 |
| **DMT1** | **28 (1.3)** | **152 (0.8)** | **1.62 (1.08,2.44)** | **0.02** | 8 (0.5) | 46 (0.4) | 1.55 (0.73,3.32) | 0.26 |
| Lupus | NR | 24 (0.1) | 0.78 (0.18,3.30) | 0.73 | 2 (0.1) | 8 (0.1) | 2.24 (0.48,10.61) | 0.31 |
| **Psoriasis** | **42 (1.9)** | **301 (1.5)** | **1.27 (0.91,1.76)** | **0.16** | 15 (1.0) | 148 (1.1) | 0.88 (0.52,1.51) | 0.65 |
| Other AI NOS[e] | 36 (1.6) | 324 (1.7) | 0.98 (0.69,1.39) | 0.92 | 20 (1.4) | 145 (1.1) | 1.27 (0.79,2.04) | 0.32 |
| Rosacea | 29 (1.3) | 277 (1.4) | 0.92 (0.63,1.36) | 0.67 | 13 (0.9) | 129 (1.0) | 0.88 (0.49,1.57) | 0.67 |
| **COPD/Emphysema** | **77 (3.5)** | **243 (1.2)** | **2.90 (2.23,3.78)** | **0.00** | **13 (0.9)** | **45 (0.3)** | **2.52 (1.34,4.71)** | **0.00** |
| **Anemia** | **128 (5.9)** | **896 (4.6)** | **1.27 (1.05,1.54)** | **0.01** | **44 (3.0)** | **283 (2.2)** | **1.40 (1.01,1.95)** | **0.04** |

*aOR* adjusted odds ratio, *CI* confidence interval, *NIGC* Non-infective gastroenteritis, and colitis, *GERD* Gastroesophageal reflux disease, *DMT1* Diabetes Mellitus Type 1, *NR* Not reportable because of CPRD cell size (<5) reporting requirements, *AI* autoimmune, *COPD* Chronic Obstructive Pulmonary Disorder.
Bolded rows have a statistical *p*-value ≤ 0.20 in a time interval.
aOR: Conditional logistic regression models [Controls were individually matched (5–10) to cases on year of birth (+/− 2 years); sex; general practice or region (general practice first, then region if we could not identify a control within the same practice); and year of practice registration (+/− 2 year)] odds ratios are adjusted for age (linear term).
[a]Includes 2188 never-smoking lung cancer cases and 19,597 never-smoking controls with at least 1 year of registration within a CPRD-Aurum primary care clinic.
[b]Includes 1455 never-smoking lung cancer cases and 12,931 never-smoking controls with at least 10 years of registration within a CPRD-Aurum primary care clinic.
[c]Consists of terms indicative of bronchitis, tracheitis, or upper respiratory infections.
[d]Consists of terms indicative of peptic, gastric, or duodenal ulcers.
[e]Other autoimmune conditions not classified.
Multiple testing correction is not needed because these are validation analyses.

bronchitis and tracheitis (aOR$_{1-10 \text{ years prior}}$ = 1.25, 95%CI 1.09–1.43); psoriasis (aOR$_{1-10 \text{ years prior}}$ = 1.29, 95%CI 1.02–1.62); diabetes mellitus type 1 (DMT1: aOR$_{1-10 \text{ years prior}}$ = 1.74, 95%CI 1.05–2.89); other autoimmune conditions NOS (aOR$_{1-10 \text{ years prior}}$ = 1.77, 95%CI 1.24–2.53); and rosacea (aOR$_{1-10 \text{ years prior}}$ = 1.52, 95%CI 1.09–2.12). Of these 12 sub-categories, COPD (aOR$_{10-32 \text{ years prior}}$ = 3.28, 95%CI 1.59–6.80), lupus (aOR$_{10-32 \text{ years prior}}$ = 3.30, 95%CI 0.89–12.31), and gastritis/NIGC (aOR$_{10-32 \text{ years prior}}$ = 1.53, 0.96–2.43) were similarly associated with LCINS in the 10–32 years prior to the index date (Table 3, Fig. 1 and Supplementary Fig. S1).

## Validation

We identified 2188 LCINS cases and matched 19,597 never-smoking controls who had been enrolled in a CPRD-Aurum general practice for a median of 15.7 years (IQR 7.3–27.5) prior to the index date (Table 1). See Source Data file S1_ValidationAurumCodes for codes used to identify medical conditions. Compared with the CPRD-GOLD population, the CPRD-Aurum population was more likely to have index dates in later years.

Conditions which were significantly associated (with *p* < 0.05) with LCINS risk in the validation dataset as in the discovery dataset in either time period included: COPD (aOR$_{1-10 \text{ years prior}}$ = 2.90, 95%CI 2.23–3.78; aOR$_{10-32 \text{ years prior}}$ = 2.52, 95%CI 1.34–4.71); upper respiratory infections (aOR$_{1-10 \text{ years prior}}$ = 1.30, 95%CI 1.16–1.46; aOR$_{10-32 \text{ years prior}}$ = 1.20, 95%CI 1.01–1.42); DMT1 (aOR$_{1-10 \text{ years}}$ = 1.62, 1.08–2.44; aOR$_{10-32 \text{ years}}$ = 1.55, 95%CI 0.73–3.32); anemia (aOR$_{1-10 \text{ years}}$ = 1.27, 95% CI 1.05–1.54; aOR$_{10-32 \text{ years}}$ = 1.40, 95%CI 1.01–1.95); GERD (aOR$_{1-10 \text{ years}}$ = 1.32, 1.09–1.60; aOR$_{10-32 \text{ years}}$ = 1.11, 95%CI 0.78–1.58); and gastritis/NIGC (aOR$_{1-10 \text{ years}}$ = 1.14, 95%CI 0.84–1.53; aOR$_{10-32 \text{ years}}$ = 1.64, 95%CI 1.08–2.49). Conditions suggestively associated include tuberculosis (aOR$_{1-10 \text{ years}}$ = 2.01, 95%CI 0.97–4.18; aOR$_{10-32 \text{ years}}$ = 2.44, 95% CI 0.81–7.35) and psoriasis (aOR$_{1-10 \text{ years}}$ = 1.27, 95%CI 0.91–1.76; aOR$_{10-32 \text{ years}}$ = 0.88, 95%CI 0.52–1.51) (Table 4, Fig. 1 and Supplementary Fig. 1).

**Sensitivity analyses.** In addition adjusting models for both body mass index (BMI) and socioeconomic status (SES) did not materially alter the associations seen in CPRD-Aurum (Table 5) or CPRD-GOLD data (Supplementary Table S2). We adjusted for medication use (ever use) in both the discovery and validation analyses for COPD, GERD, and gastritis/NIGC. We only adjusted for medication use for psoriasis, lupus, and rosacea in CPRD-GOLD, because they were not statistically associated with LCINS in CPRD-Aurum. We examined use of the following medications with codes available in both CPRD-GOLD and CPRD-Aurum: oral corticosteroids; inhaled corticosteroids; methotrexate; other immunosuppressants; Long-Acting Beta-2 Agonists (LABA); Short-Acting Beta-2 Agonists (SABA); Antimuscarinic Bronchodilators; macrolides; proton pump inhibitors (PPI); H2-receptor antagonists; and antacids; Nonsteroidal Anti-Inflammatory Drugs (NSAIDs); and tetracyclines. Associations between LCINS and the conditions were only modestly attenuated when medications were added to the statistical models (Table 6).

Of note, iron-related anemia appeared to drive the anemia associations with LCINS (Supplementary Table S3). However, larger studies are needed to confirm these findings.

## Discussion

In this large, population-based study of PWHNS, we used EMRs to agnostically identify and independently validate medical conditions potentially associated with LCINS risk. To the best of our knowledge, this is the largest study of LCINS risk[17]. Our approach identified several putative or established risk factors, and some conditions without prior associations with LC.

Among the previously reported risk factors, the strongest and most consistent association observed in this study was for COPD/emphysema. Although COPD is generally associated with smoking, a recent review estimates that >50% of all cases globally occur in PWHNS, with estimates varying geographically and by SES[18]. Risk

**Table 5 | Associations between medical conditions and LCINS in the validation (CPRD-Aurum) dataset 1–10 years before selection adjusted for body mass index (BMI) and socioeconomic status (SES, index of multiple deprivation)**

| | No. (%) in cases | No. (%) in controls | aOR (95% CI) | aOR1 (95% CI) |
|---|---|---|---|---|
| **Infections and Inflammation** | | | | |
| **Tuberculosis** | 9 (0.4) | 41 (0.2) | 2.01 (0.97,4.18) | 2.04 (0.98,4.25) |
| Influenza | 28 (1.3) | 207 (1.1) | 1.22 (0.81,1.82) | 1.21 (0.81,1.81) |
| **Upper Respiratory Infections**[c] | 444 (20.3) | 3263 (16.7) | 1.30 (1.16,1.46) | 1.29 (1.15,1.45) |
| **Gastritis/NIGC** | 50 (2.3) | 392 (2.0) | 1.14 (0.84,1.53) | 1.12 (0.83,1.51) |
| **GERD** | 132 (6.0) | 917 (4.7) | 1.32 (1.09,1.60) | 1.30 (1.07,1.57) |
| Gastrointestinal Ulcers[d] | 30 (1.4) | 209 (1.1) | 1.23 (0.84,1.81) | 1.23 (0.84,1.81) |
| **DMT1** | 28 (1.3) | 152 (0.8) | 1.62 (1.08,2.44) | 1.60 (1.06,2.41) |
| Lupus | NR | 24 (0.1) | 0.78 (0.18,3.30) | 0.75 (0.18,3.18) |
| **Psoriasis** | 42 (1.9) | 301 (1.5) | 1.27 (0.91,1.76) | 1.26 (0.91,1.75) |
| Other AI not classified[e] | 36 (1.6) | 324 (1.7) | 0.98 (0.69,1.39) | 0.96 (0.68,1.37) |
| Rosacea | 29 (1.3) | 277 (1.4) | 0.92 (0.63,1.36) | 0.93 (0.63,1.37) |
| **COPD/Emphysema** | 77 (3.5) | 243 (1.2) | 2.90 (2.23,3.78) | 2.83 (2.17,3.69) |
| **Anemia** | 128 (5.9) | 896 (4.6) | 1.27 (1.05,1.54) | 1.26 (1.04,1.53) |

*aOR* adjusted odds ratio, *CI* confidence interval, *NIGC* Non-infective inflammatory gastroenteritis and colitis, *GERD* Gastroesophageal reflux disease, DMT1, Diabetes Mellitus Type 1, *NR* Not reportable because of CPRD cell size (< 5) reporting requirements, *AI* autoimmune; *COPD,* Chronic Obstructive Pulmonary Disorder.
Bolded rows were also bolded in Table 4.
aOR: Conditional regression model [Controls were individually matched to cases on year of birth (+/− 2 years); sex; general practice or region (general practice first, then region if we could not identify a control within the same practice); and year of practice registration (+/− 2 year)] odds ratios are adjusted for age (linear term). They are the same associations as those presented in Table 4 (1–10 years).
aOR1: Further adjusted for body mass index (kg/m2) [< 18.5, 18.5–24.9, 25.0–29.9,≥ 30.0, missing or recorded > 15 years before selection] and index of multiple deprivation in deciles fitted with a trend.
[a]Includes 2188 never-smoking lung cancer cases and 19,597 never-smoking controls with at least 1 year of registration within a CPRD-Aurum primary care clinic.
[c]Consists of terms indicative of bronchitis, tracheitis, or upper respiratory infections.
[d]Consists of terms indicative of peptic, gastric, or duodenal ulcers.
[e]Other autoimmune conditions not classified.

factors include childhood respiratory diseases, air pollution, occupational exposures[18,19] as well as dysanapsis, a mismatch of airway tree caliber to lung size[20]. Several cytokines (e.g., interleukin (IL)-8, IL-6, IL-1β) that are elevated in persons with COPD have been previously associated with LC risk (including LCINS) and are important for LC and/or cancer biology[21–25]. Our associations are also similar to those reported in a recent retrospective cohort study from South Korea (hazard ratio [HR] = 2.67, 95%CI 2.09–3.40)[26]. Adjusting for several of the medications potentially used in the treatment of COPD modestly attenuated the association, suggesting that they might mediate some of the association with COPD.

Another previously reported association that we validated was upper respiratory infections, a category largely comprised of bronchitis[12,27]. Tuberculosis, which was statistically significant in the discovery phase, was not statistically significant in the validation phase (*p* = 0.06). Pneumonia has commonly been associated with LC[12], although it was not statistically significantly associated with LCINS in this study in the discovery phase. Asthma was also not associated with LCINS in this study. However, the UK Million Woman Study found that only asthma requiring treatment was associated with LCINS risk in women[28].

We validated several conditions with little to no previous evidence supporting a relationship with LC including DMT1, anemia, gastritis, and GERD. There is conflicting evidence of an association between diabetes mellitus (DM; both types) and LC[29], but the use of Metformin, an antidiabetic medication, has been associated with reduced LC risk[30]. Further, individuals with DM are at increased risk of lung diseases like asthma, COPD, and pulmonary fibrosis[31]. Interestingly, IL-21 deficiency, which can lead to immunosuppression, is dysregulated in several autoimmune conditions including DMT1, and was inversely associated with LCINS risk in a nested case-control study of women[23,32].

The association between anemia and LCINS could reflect the presence of latent cancer, although it was associated with LCINS 10-32 years prior to the index date. Furthermore, our results are similar to those from a study of iron deficiency anemia and LC in a cohort of Taiwanese healthcare beneficiaries. The authors speculated that anemia may render the microenvironment conducive to carcinogenesis[33]. Biologically, inflammation-related anemia can be caused by the pro-inflammatory cytokine IL-6, which regulates serum iron and has been associated with elevated LC risk (including PWHNS)[23,34]. Interestingly, the association between iron-related anemia SNOMED codes and LCINS was similar to the association between anemia overall and LCINS. Furthermore, both diabetes[35] and gastritis[36] can cause anemia.

To the best of our knowledge, there are no studies supporting an association between gastritis and LCINS, and the single study reports an association (HR = 1.53 95%CI = 1.19–1.98) between GERD and LC combined smokers and non-smokers[37]. GERD is one of the leading causes of chronic cough and is common in several inflammatory lung diseases such as cystic fibrosis, asthma, and idiopathic pulmonary fibrosis (IPF)[38]. It has been suggested that GERD might play a role in their pathogenesis, perhaps by causing chronic microaspiration[39–41]. Although most individuals with GERD do not experience chronic microaspiration, most persons with occult chronic aspirations have GERD[42]. Interestingly, hernia, which was associated with LCINS in the discovery stage—albeit not meeting the multiple testing correction (aOR = 1.29, 95%CI 1.06–1.56, *p* = 0.01)— can cause GERD[43].

The associations were slightly attenuated for both GERD and gastritis when we adjusted for the use of PPIs or H2 receptor antagonists which are common treatments. PPIs, H2 receptor antagonists, and antacids were independently associated with LCINS risk. PPI use has been linked to various cancers, including gastric cancer in the CPRD[44]. Several mechanisms by which PPIs might contribute to cancer development have been proposed, including alteration of the gut microbiome which can affect inflammation, immune responses, and the production of carcinogenic metabolites by overgrowth of harmful bacteria. PPI-induced hypergastrinemia can cause proliferation and hyperfunction of enterochromaffin-like (ECL) cells[45]. Our observations could also be driven by vague symptoms that are due to a latent, yet

**Table 6 | Associations between medical conditions and LCINS in both datasets adjusted for ever use of medications**

| Condition | Medication(s) | aOR$_{Unadj}$ (95%CI) | aOR$_{Cond}$ (95%CI) | aOR$_{Med}$ (95%CI) |
|---|---|---|---|---|
| **Population: CPRD-GOLD** | | | | |
| COPD | Oral Corticosteroids | 2.91 (2.12,3.99) | 2.42 (1.74,3.35) | 1.44 (1.22,1.70) |
| COPD | LABA | 2.91 (2.12,3.99) | 2.64 (1.88,3.71) | 1.25 (0.95,1.65) |
| COPD | SABA | 2.91 (2.12,3.99) | 2.60 (1.86,3.63) | 1.18 (1.00,1.39) |
| COPD | Inhaled Corticosteroids | 2.91 (2.12,3.99) | 2.84 (2.02,4.00) | 1.04 (0.85,1.26) |
| COPD | Antimuscarinic Bronchodilators | 2.91 (2.12,3.99) | 2.19 (1.50,3.19) | 1.76 (1.20,2.57) |
| COPD | Macrolides | 2.91 (2.12,3.99) | 2.87 (2.09,3.94) | 1.07 (0.93,1.24) |
| GERD | PPIs | 1.39 (1.11,1.74) | 1.24 (0.97,1.57) | 1.23 (1.08,1.40) |
| GERD | H2 Receptor Antagonists | 1.39 (1.11,1.74) | 1.31 (1.04,1.65) | 1.31 (1.10,1.56) |
| GERD | Antacid | 1.39 (1.11,1.74) | 1.27 (1.01,1.61) | 1.32 (1.12,1.55) |
| Gastritis/NIGC | PPIs | 1.56 (1.13,2.16) | 1.41 (1.01,1.96) | 1.25 (1.10,1.41) |
| Gastritis/NIGC | H2 Receptor Antagonists | 1.56 (1.13,2.16) | 1.45 (1.04,2.02) | 1.32 (1.11,1.57) |
| Gastritis/NIGC | Antacid | 1.56 (1.13,2.16) | 1.45 (1.05,2.02) | 1.34 (1.14,1.57) |
| Psoriasis | Methotrexate | 1.35 (0.91,2.00) | 1.33 (0.90,1.97) | 1.60 (0.90,2.84) |
| Lupus | Immunosupressing | 4.54 (1.55,13.28) | 3.84 (1.28,11.47) | 2.32 (1.55,3.49) |
| Lupus | Oral Corticosteroids | 4.54 (1.55,13.28) | 3.93 (1.33,11.57) | 1.57 (1.34,1.85) |
| Lupus | NSAIDs | 4.54 (1.55,13.28) | 4.55 (1.55,13.32) | 0.99 (0.88,1.11) |
| Rosacea | Macrolides | 1.44 (0.99,2.10) | 1.42 (0.97,2.07) | 1.10 (0.96,1.27) |
| Rosacea | Tetracyclines | 1.44 (0.99,2.10) | 1.32 (0.90,1.93) | 1.23 (1.04,1.45) |
| **Population: CPRD-Aurum** | | | | |
| COPD | Oral Corticosteroids | 2.90 (2.23,3.78) | 2.66 (2.03,3.49) | 1.22 (1.07,1.40) |
| COPD | Antimuscarinic Bronchodilators | 2.90 (2.23,3.78) | 2.10 (1.54,2.86) | 2.00 (1.47,2.71) |
| COPD | SABA | 2.90 (2.23,3.78) | 2.36 (1.79,3.12) | 1.36 (1.19,1.54) |
| COPD | LABA | 2.90 (2.23,3.78) | 2.45 (1.85,3.26) | 1.48 (1.18,1.84) |
| GERD | Antacid | 1.32 (1.09,1.59) | 1.24 (1.02,1.51) | 1.23 (1.08,1.40) |
| GERD | PPIs | 1.32 (1.09,1.59) | 1.21 (0.99,1.48) | 1.16 (1.04,1.29) |
| GERD | H2-receptor Antagonists | 1.32 (1.09,1.59) | 1.24 (1.02,1.51) | 1.32 (1.15,1.52) |
| Gastritis/NIGGC | Antacid | 1.14 (0.84,1.53) | 1.07 (0.79,1.45) | 1.26 (1.10,1.43) |
| Gastritis/NIGGC | PPIs | 1.14 (0.84,1.53) | 1.05 (0.77,1.42) | 1.19 (1.07,1.32) |
| Gastritis/NIGGC | H2-receptor Antagonists | 1.14 (0.84,1.53) | 1.06 (0.78,1.43) | 1.35 (1.17,1.55) |

*aOR* adjusted odds ratio, *CI* confidence interval, *COPD* Chronic Obstructive Pulmonary Disorder, *GERD* Gastroesophageal reflux disease, *NIGC* Non-infective gastroenteritis and colitis
Medication Abbreviations: *LABA*, Long Acting Beta-2 Agonists; *SABA*, Short Acting Beta-2 Agonists; *PPIs*, Proton Pump Inhibitors; *NSAIDs*, Nonsteroidal Anti-Inflammatory Drugs.
All medication Read and Snomed codes are listed in discoveryMedications.xlsx and ValidationMedications.xlsx.
aOR$_{Unadj}$: Associations between condition (first column) as documented in 1–10 years before selection and LCINS. Conditional regression models are adjusted for age (linear term). Controls were individually matched to cases on year of birth (+/− 2 years); sex; general practice or region (general practice first, then region if we could not identify a control within the same practice); and year of practice registration (+/− 2 year).
aOR$_{Cond}$: Same as aORUnad but the statistical model also includes if a prescription for the medication(s) listed in column 2 was observed in the clinical history in the 1–10 years before selection.
aOR$_{Med}$: Association between the medication(s) as documented in the 1–10 years before selection (second column) and LCINS adjusted for age (linear term) and the condition listed in the first column. The statistical model is identical to that described above (aOR$_{Cond}$).

undetected LCINS, which led to antacid or PPI/H2 receptor antagonist use.

The suggestive association between psoriasis and LCINS in CPRD-GOLD was similar in magnitude in CPRD-Aurum, albeit not statistically significant. A single study that adjusted for smoking status reported that only severe psoriasis was associated with LC[46]. Severe psoriasis is treated with methotrexate which can cause pulmonary fibrosis[47,48]. The association between psoriasis and LCINS was modestly attenuated when methotrexate use was added to the statistical model. Psoriasis is also associated with asthma[49].

Unfortunately, we could not examine the association between IPF and LCINS[50] as we did not identify any IPF from LCINS cases included in our feasibility study. This was expected because the prevalence of IPF in the UK is estimated at only 0.78 per 10,000 persons (0.38, 1.63)[51].

Our study has some limitations. First, our grouping of medical conditions was based on our clinical expertize interpreting clinical terminologies. The coding and definitions of some diseases are likely to vary across different doctors, general practices, and calendar years.

Further, some conditions (e.g., anemia and gastritis) have multiple etiologies and our study was not designed to differentiate between heterogeneous entities. This study also did not account for disease severity and/or manifestations. For example, the elevated risk for lupus in CPRD-GOLD that was not replicated in CPRD-Aurum may have been driven by systemic lupus erythematosus (SLE) which can affect the lungs and is treated by immunosuppressant medications such as azathioprine. SLE is more common in Northern Ireland and Scotland, two regions not represented in the validation dataset[52]. While we examined the impact of adjusting for medications on condition-LCINS associations, we relied on only single medication codes (ever exposure), although most medications examined are expected to be used on a more chronic basis. Further, data on some medication use may be unavailable in CPRD (e.g., medications administered in a hospital setting or prescribed by a specialist). However, this study was not designed to robustly examine specific medication-LCINS associations and these findings should be interpreted cautiously.

There are additional limitations. The medical ontology used in the discovery stage differed from that used in the validation stage which

may explain why some conditions did not validate. Also, our study was not linked to cancer registries. Only ~ 50% of individuals in the CPRD-GOLD are linkable to cancer registries, and that restriction would have greatly reduced sample size. However, there is high concordance between cancers identified in the CPRD and those found in UK cancer registries[53]. Finally, this study was also not designed to examine how individual LCINS risk factors jointly influence LCINS risk and several of the conditions co-occur (e.g., autoimmune disease and anemia). Cancer risk factors rarely act alone and future, more targeted studies need to examine the interplay between the conditions and medications in relation to LCINS risk. The study limitations should be viewed within the goals of this hypothesis-generating study which was to identify etiologic clues to LCINS.

Our study had several strengths. The large study population had a median of 15 years of clinical history available. The CPRD-GOLD dataset is research quality and contains smoking status on ~ 95% of the population, and recorded smoking status has high validity[54,55]. The identification and validation of associations in expected conditions (e.g., COPD, tuberculosis) gives us confidence in our agnostic approach. Finally, we employed a conservative multiple-testing correction and validated our results in an independent population.

Most of the medical conditions we identified involved pulmonary or systemic inflammation. Furthermore, several of the associations persisted for >10 years, hinting at the importance of long-term inflammation in the development of LCINS. The identification of medical conditions that were present many years before the LC diagnosis in this study agrees with the findings of a recent study from the Sherlock-*Lung* project that identified three molecular subtypes of LCINS[16]. That study reported that a large proportion of these cancers had a long latency of up to a decade before diagnosis, providing a window of time for early detection. Moreover, these cancers had specific genomic alterations suggestive of stem cells that had exited their quiescent state; inflammation (as in the medical conditions we identified here) can cause tissue damage and consequent tissue regeneration with stimulation of stem cells.

Our results are particularly relevant because of the inflammatory lung damage that SARS-CoV-2 can cause, with evidence that extended post-infection symptoms (long COVID) is associated with a two-fold increased risk for COPD, severe asthma, and pulmonary fibrosis[56,57]. In addition, air pollution which is associated with an increased risk of COPD in non-smokers[18], has been shown to promote lung adenocarcinoma by causing a release of IL-1β, a pro-inflammatory cytokine that plays a role in the pathogenesis of LC and COPD[25,58].

The potentially long latency period might provide an opportunity to identify individuals who would benefit from earlier LC detection. The growing burden of LCINS requires developing strategies that are not reliant upon smoking history for LC risk assessment in LC screening ineligible persons. Our findings highlight the potential utility of using routinely collected data from clinical practice to identify signals associated with elevated LCINS risk.

## Methods
### Ethics
This study uses data from the CPRD, obtained under license from the U.K. Medicines and Healthcare Products Regulatory Agency. The data collected by the National Health Service (NHS) as part of routine care, are provided by patients. The CPRD Independent Scientific Advisory Committee reviewed and approved this study (proposal #18_160.RAR). Since the National Cancer Institute only received de-identified data from CPRD, had no direct contact or interaction with the study participants, and did not use or generate identifiable private information, Sherlock-Lung has been determined to constitute 'non-human subject research' based on the federal Common Rule (45 CFR 46; https://www.ecfr.gov/cgi-bin/ECFR?page=browse).

### Study design
We employed a two-stage approach. In the first stage, we identified conditions associated with LCINS risk using CPRD-GOLD. Then we validated these conditions in an independent dataset, CPRD-Aurum.

### Study design for the discovery stage
We performed a nested case-control study using the UK's CPRD[54] GOLD database. CPRD-GOLD is a research-quality, population-based database established in 1987. It currently collects EMRs from 985 primary care practices, covering ~ 20 million UK residents[59]. CPRD is the world's largest computerized database of anonymized longitudinal patient records from primary care practices and CPRD-GOLD is sex and age representative of the UK population[54]. It includes demographic characteristics, clinical diagnoses, referral information, specialty consultation notes, laboratory test results, and prescriptions. Importantly, smoking status is available for > 95% of individuals, as primary care physicians are paid to inquire about smoking status and then update that information if necessary[54,60]. Clinical events are recorded using Read codes[61].

### Discovery population
We identified all invasive primary first lung cancer cases diagnosed Jan 1, 1988 - Dec 31, 2019 (n = 77,099). We excluded cases with less than one year of registration prior to diagnosis in a CPRD-GOLD general practice (n = 13,731), or any evidence of any invasive cancer (except basal cell carcinoma and cervical cancer) before diagnosis (n = 10,867). Unlike cutaneous squamous cell carcinoma, basal cell carcinoma does not metastasize to the lung. Cervical cancer was not excluded so that we could test the hypothesis that the human papilloma virus is associated with lung cancer[62]. Unfortunately, the number of cervical cancer cases was too low (n = 2) for this purpose. We excluded cases with no documented smoking status in the full clinical history or any evidence of smoking in either Read codes or entity codes from practice visits (n = 50,643). Controls who had never smoked with at least one year of general practice registration and no evidence of a prior cancer were identified. Between 5 and 10 controls who were cancer-free, alive, and enrolled in CPRD-GOLD at case diagnosis date (termed selection date for controls) were individually matched to cases on year of birth (+/− 2 years); sex; general practice or region (general practice first, then region if we could not identify a control within the same practice because regions are large); and year of practice registration (+/− 2 year). We further excluded cases with less than five matched controls (n = 28), cases diagnosed before 30 or after 89 years of age (n = 129), and cases with evidence of smoking after LC diagnosis (n = 120), resulting in 1581 cases included in our discovery analyses.

### Exposure classification for the discovery stage
A feasibility study conducted in late 2018 in the CPRD-GOLD dataset yielded 1478 LCINS cases, from which we extracted all Read codes (indicating diagnoses) present in the clinical history. We grouped case Read codes into clinically meaningful and specific disease categories. We required each disease category (called subcategory below) to include at least 15 cases, corresponding to a case prevalence of ~ 1% (0.01 × 1478 LCINS = ~ 15). This was based on statistical power calculations under a range of reasonable assumptions, including a multiple testing corrected alpha level (Supplementary Table S4). We supplemented our conditions with codes already generated within our Division for prior CPRD studies (Source data file S2_DiscoveryGOLDCodes).

These categories were then further restructured into 24 primary categories, in total containing 98 distinct subcategories (e.g., upper respiratory conditions within the infections and inflammation category). Some primary categories, such as infections and inflammation had many subcategories, whereas others (e.g., anemia) had none. This strategy was used to improve statistical power by lessening the

multiple testing burden and to enable the fitting of statistical hierarchical models. See Supplementary Table S5 for the hierarchy of conditions. We excluded codes that were not related to a specific disease or suspected disease. Nor did we consider conditions such as obesity or alcoholism, given that they are not uniformly collected across the practices. Moreover, excessive alcohol consumption in the UK might be associated with passive smoking because alcohol consumption commonly occurred in pubs where smoking was allowed until 2007. Read codes referring to the active management of a disease, for example, management of COPD, were included.

For each condition, we identified the earliest date at which the condition was observed in the 1–10 and 10–32 years prior to the index date. Events identified within the year before the index date were excluded to reduce the potential for reverse causation. Primary analyses were performed for diagnoses occurring in the 1–10 year interval, because it contained the largest sample size (individuals only had to be registered for one year) and it is more representative of the UK population. The 1–10 year exposure interval is relevant also because genomic results show that LCINS may progress from the progenitor cells a decade before diagnosis. We explored whether the associations we identified in the 1–10 year assessment period persisted 10–32 years prior to the index date. Diagnoses before Jan 1,1987 (32 years before 2019) were left truncated as the data are not considered research standards before this date.

## Statistical analyses
We used a two-stage approach to identify medical conditions associated with LCINS risk in the 1–10 years before the index date. First, we estimated adjusted odds ratios (aORs) and 95% confidence intervals (CIs) for all 24 primary disease categories (coded as 1 if any condition in that category was present in the clinical history 1–10 years prior to the index date and 0 otherwise) using conditional logistic regression models to account for the matched design. We additionally adjusted models for age at index date in single years. For each primary disease category that was associated with LCINS at an FDR $p$-value < 0.05, we then fit a hierarchical (random effects) logistic regression model (second stage) that included all individual conditions in that primary category. Under that model, the aORs of the individual conditions that comprise the primary category are assumed to vary randomly around the mean of the primary category (details are given in **Supplementary methods: Hierarchical analyses**). We adjusted the hierarchical models for matching factors to accommodate the matched study design. Because the aORs of the individual conditions are mutually adjusted for all other conditions within the primary category under the hierarchical model, a multiple testing correction is not needed, and $p < 0.05$ was considered statistically significant. Secondary analyses were performed for conditions diagnosed 10–32 years prior to the index date to explore associations with longer lag periods.

## Sensitivity analyses
To assess the robustness of the hierarchical regression model results, we examined associations between all primary categories and individual conditions in both time intervals and LCINS risk using conditional logistic regression. Conditioning on the matching variables removes the effect of any unmeasured confounders correlated with the matching variables and thus is a more robust, although less efficient statistical analytic approach. The aORs from conditional logistic models are not adjusted for other individual conditions in the same primary disease category.

We used conditional logistic regression to examine the 1–10 year condition-LCINS associations additionally adjusted for BMI and SES for those conditions that were statistically significant in the discovery stage. SES is associated with many diseases and has been shown to be independently associated with LC risk[63]. SES was mapped to deciles of social deprivation and included in the models as a linear term for

individuals with available linkage to the CPRD index of multiple deprivation[64]. In CPRD-GOLD, 43% of the study population had a linkage to SES information. BMI, which has been more consistently recorded over time[65], was available for ~75% of the study population. The most recent BMI measurements were used. Measurements occurring >15 years before the index date were set to missing.

We also examined if medications used to treat particular conditions might modify the association estimates of the conditions on LCINS risk[66] (see list of medications in Source Data file S3_MedicationsGOLD). We present associations from conditional logistical regression models which include both the condition and the medication(s) that may have been used to treat the condition. UK treatment guidance and the British National Formulary (September 2020-March 2021) were used to compile a list of medications used in the treatment of conditions identified as being associated with LCINS. Medications were then identified from the records of LCINS cases. We required only a single medication prescription within 1–10 years for an individual to be considered exposed to the medication (ever use).

## Independent validation in CPRD-Aurum
Specific conditions (not primary disease categories) that were significantly associated ($p < 0.05$) with LCINS in CPRD-GOLD were validated in the independent CPRD-Aurum dataset. CPRD-Aurum is a population-based research quality database covering 10 regions of England[59]. Case and control selection was identical to the discovery stage. Among 105,679 lung cancer cases diagnosed between Jan 1, 1988 and Dec 31, 2019, we excluded cases with less than one year of registration prior to diagnosis ($n = 2737$), evidence of invasive cancer (except basal cell carcinoma and cervical cancer) before diagnosis ($n = 33,629$), evidence of smoking or no documented smoking status according to the full clinical history ($n = 65,984$), cases with less than five controls ($n = 36$), cases diagnosed before 30 or after 89 ($n = 247$), and cases with any evidence of smoking after diagnosis ($n = 206$). Individuals who were present in both databases (23%) were also excluded from the validation dataset ($n = 652$). After exclusions, the validation dataset included 2188 cases. We removed controls who lost matched cases in the deduplication process. Unlike CPRD-GOLD, CPRD-Aurum does not capture patients in Northern Ireland, Scotland, or Wales and is therefore not geographically representative of the entire UK population[67]. For each validated condition, we identified the earliest date at which the condition was observed in the 1–10 and 10–32 years prior to the index date in CPRD-Aurum. Conditions only identified within the year before the index date were excluded to reduce the potential for reverse causation. SNOMED codes were used to classify conditions in CPRD-Aurum. Conditional logistic regression models were used to estimate aORs between conditions and LCINS for both the 1–10 and 10–32 year assessment windows. We present all associations but focus on those that were significant ($p < 0.05$) in CPRD-Aurum.

As in the discovery stage, we also performed sensitivity analyses in which conditional regression models were jointly adjusted for BMI and SES for significant findings. In contrast to the CPRD-GOLD population, most of the CPRD-Aurum population is linkable to SES information, thereby providing a more robust interpretation of the results. Following a similar strategy in the discovery stage, we also adjusted for medication use in statistical models for conditions associated with LCINS. See Source Data file S4_MedicationsAurum for the list of medications. For select validated conditions that can have multiple etiologies (e.g., anemia), we attempted to use SNOMED codes to identify different manifestations/etiological subtypes of the disease and then examine associations with LCINS.

## Reporting summary
Further information on research design is available in the Nature Portfolio Reporting Summary linked to this article.

## Data availability

This study is based in part on data from the Clinical Practice Research Datalink obtained under license from the UK Medicines and Healthcare Products Regulatory Agency. The data is provided by patients and collected by the NHS as part of their care and support. The data is accessible exclusively to researchers with protocols approved by the CPRD's independent scientific advisory committee. The scientifically approved proposal and amendments of this study (18_160.RAR) and further information are available at: https://www.cprd.com/data-access. Source data are provided in this paper.

## Code availability

Programs written in R version 4.3.2 and SAS version 9.4 used to clean and analyze data are available at: github.com/jpmcelderry/NCOMMS-23-60017A-analytic-code.

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

## Acknowledgements

This work was funded by the Intramural Research Program, Division of Cancer Epidemiology and Genetics, National Cancer Institute, NIH, DHHS.

## Author contributions

M.T.L. conceptualized the study. M.T.L., M.E.D., R.M.P., C.T.D., S.M.G., M.C.B., and Y.W. designed the study. C.T.D., M.K., S.R., M.L., M.E.D., Y.W., M.C.B., and M.T.L. contributed to the curation of data and the generation of condition groupings and hierarchy. M.E.D. performed formal analyses. R.M.P. and J.P.M. revised the R and SAS codes for data cleaning and analyses. M.C.B. defined the medication list for the analyses and helped interpret results related to medication use. C.T.D., M.K., M.E.D., R.M.P., S.M.G., Y.W., S.R., and M.T.L. wrote and made amendments to the CPRD proposal. M.K., M.E.D., M.L., Y.W., S.M.G., and R.M.P. participated in making data requests to research analysts. M.E.D., M.T.L., M.C.B., S.M.G., Y.W., and R.M.P. interpreted the results. M.E.D. visualized the results. M.T.L. acquired funding for the project. R.M.P. and M.T.L. supervised the study. M.E.D. wrote the original draft. M.E.D., R.M.P., M.C.B., P.H.H., T.V.T.T., J.P.M., M.L., M.K., C.T.D., S.R., Y.W., S.M.G., and M.T.L. reviewed and edited the manuscript. M.E.D., M.K., and R.M.P. directly accessed and verified the underlying data reported in the manuscript. All authors approved the final version of the manuscript.

## Funding

## Competing interests

The authors declare no competing interests.
