## [Transparent Peer Review file · Nature Communications]

Inflammatory diseases and risk of lung cancer among individuals who have never smoked

Corresponding Author: Dr Monica Darcy

Version 0:

Reviewer comments:

Reviewer #1

(Remarks to the Author)

This is a very interesting, hypothesis-generating, case-control study in large patient and control samples from UK health databases, identifying several inflammatory disease entities as potential risk factors for Lung Cancer In Never smokers (LCINS), including discovery and validation cohorts. Although this study is novel, contains original information that are useful for clinicians and may also inform screening strategies for early detection of LCINS, it harbors important caveats that should be adequately addressed before reconsideration:

MAJOR COMMENTS:

1. The Medical Conditions identified to potentially increase the risk of LCINS are quite heterogeneous: Some of them are specific disease entities (ie Lupus, psoriasis), while some others are general medical conditions with multiple aetiologies (ie anemia, GERD) and this renders the results hard to interpret because of the risk of confounding factors. For example, anemia may be idiopathic, but also may be recorded in the context of other systemic disease entities (autoimmune, hematologic, etc). The authors should discriminate between these conditions, acknowledge the limitations of the confounding factors and should stick more to the correlations of specific disease entities, rather than general medical conditions
2. Importantly, the authors unfortunately do not take into account the use of concomitant medications administered for those disease entities, although in the Methods session they clearly state that this information regarding prescriptions was available in the database, at least in the Discovery Cohort (Line 96). This information is extremely important because medications used for many of these conditions (COPD, Lupus, psoriasis, Emphysema, Rosacea) are often immunosuppressive, and this may have contributed to the occurrence of LCINS. It is well known that cancer may arise from an impaired immunity status, especially in the absence of environmental carcinogens, such as smoking. For example, the chronic use of steroids or other immunosuppressive drugs in a non-smoking patient with COPD/asthma or an autoimmune disease can cause severe chronic immunosuppression that can lead to carcinogenesis. In other words, it might be the pharmaceutical immunosuppression that causes LCINS and NOT the autoimmune or inflammatory disease per se. This is an inherent bias of the study that should be adequately addressed and, if authors are not able to adjust for this confounding variable, they should clearly state this important limitation in the Discussion session
3. The title of the Manuscript can be misleading, leading to a misinterpretation that all kinds of inflammations are associated with LCINS. I would suggest to modify the title as "Inflammatory diseases and risk of lung cancer among never smokers" or something similar
4. Finally, this manuscript contains high-magnitude statistical analysis and this Reviewer is not competent to perform accurate review of the Statistical Methods session. I would therefore advise Official Statistical Review of the study by a statistician

MINOR COMMENTS:

1. Line 107: What was the rate of discrepancy between the smoking codes?
2. Line 112: What about other HPV-related cancers? (Oropharyngeal, anal, etc)
3. Line 132: Why was obesity excluded from the analysis as a potential risk factor for LCINS? Please provide justification as you did for alcohol consumption (exposure to passive smoking)
4. Line 267: How do authors explain that there were no cases of recorded IPF in thousands of cases and controls studied? Is it possible that this disease code was not captured as a disease entity in the databases used?

(Remarks on code availability)

this manuscript contains high-magnitude statistical analysis and this Reviewer is not competent to perform accurate review of the Statistical Methods session. I would therefore advise Official Statistical Review of the study by a statistician

Reviewer #2

(Remarks to the Author)

This report presents the results from a comprehensive, though exploratory, analysis of EHR data compiled from two substantial databases, to examine associations of various comorbidities with lung cancer in never smokers (LCINS). The discovery dataset covers 41 primary care practices within the United Kingdom's Clinical Practice Research Datalink, called CPRD-GOLD, comprising >20 million residents, and the validation dataset is a similar but independent dataset called CPRD-Aurum, covering selected regions within the UK and 40 million residents. The authors examined a large number of conditions, some considered to have established or putative associations with LCINS. This is a methodologically rigorous analysis, with multiple sensitivity analyses and statistical approaches taken to mitigate possible biases and false discoveries. The results confirm previously established associations, and provides additional data points for putative risk factors. The newly identified risk factors are considered hypothesis generating findings. I had just two questions that the author might address.

1. The study design was a nested case-control analysis, and was robustly attentive to most possible sources of bias inherent to the data. The authors may want to comment on why they did not conduct a longitudinal time-to-event analysis using all records.

2. As the authors themselves explain, LCINS has a strong association with socioeconomic status. To the extent that some of the conditions are also associated with socioeconomic status, the authors should comment on whether it is possible that some of the associations may be due to unmeasured confounding by SES. Is it possible to adjust for some indicator of SES in these analyses?

(Remarks on code availability)

I am not a statistician nor programmer, so did not review the code.

Reviewer #3

(Remarks to the Author)

The manuscript entitled "Inflammatory diseases are associated with lung cancer in never smokers" is a hypothesis-generating study trying to identify risk factors for LCINS. Remarkably, the authors presented other recent studies supporting their results, suggesting that inflammatory diseases including COPD or GERD would be parts of LC pathogenesis. This work could broaden a perspective to establish screening programs for LC early detection. It is well-written overall, however, I have a few concerns mainly on the statistical analyses.

Line 121 explained the study conducted in 2018 however the analysis was done additionally with 2019 data as stated in Line 182. Unless there are reasons why it has to be, it'd be good to introduce exposure classification with the full data explored.

Please elaborate more on the arguments about 'statistical power' (Lines 125, 129). What was the authors' statistical power? How do the case prevalence and the categorizing strategy relate to the power?

Line 132 stated that obesity or alcoholism conditions were not considered. I understand that alcoholism-related conditions were excluded because alcohol consumption might be associated with passive smoking as the authors explained. However, what is the reasoning to exclude obesity-related conditions?

The authors called the study using individuals in the 1-10 year period prior to selection as a primary study because it has a large sample and is more representative of the UK population. It is no wonder that the larger sample size is favorable for the higher power, however, how come it is more representative?

The manuscript would be enhanced if the two-stage structure of the primary (conditional logistic) and secondary (hierarchical; logistic) models were more clearly stated. Currently, terminologies are mixed so it wasn't easy to understand how the models were fitted. What is "this strategy" in Line 128 that enables the fitting of statistical hierarchical models? Does the "two-phase" approach indicate the primary-secondary structure based on broader/sub-categories, or 1-10 year plus 10-32 year analysis to explore whether the associations identified in 1-10 years persisted more than 10 years? What is the motivation of including random effects in which subcategories vary randomly around the mean of the primary category?

What does it mean that the ORs of second stage model are “mutually adjusted”, and how come does it lead to $p < 0.05$ being considered as significant? Why the matching factors are adjusted in the hierarchical model even though the data are from a nested case-control study? Why subcategories (peripheral vascular and myocardial infarction) were fitted even though its primary category (cardiovascular) was not strong enough, which contradicts the explanations in Lines 153-155? More elaboration on mathematical representations in the appendix and/or relevant references would help for this purpose.

I wonder for what purpose the sensitivity analyses were performed and how. (Line 160)

The authors said the validation set does not capture some regions so that it is not geographically representative (Line 169). Can this explain any of the different result from the discovery study, such as the opposite direction of Lupus in Fig 1?

It is unclear in Lines 289-291 that 1% COPD in the controls supports the good quality of smoking assessment.

The Tables 2 and 4 are not fully shown. Please update the right columns that show p-values and/or FDR.

In the tables, NR stands for “not reported” due to small cell size (< 5) however there are cells stating “ < 5 ” instead of NR. Please clean up the notations.

Please clarify how the validation was performed so that it is persuasive. Was the primary model for the validation set fitted but not reported, or not fitted but the secondary model can be fitted based on the discovery set analysis? Which stage is for “...used conditional logistic regression... in the validation phase” in Line 172 (if the second, is it the right way)?

(Remarks on code availability)

I am willing to review the code once the authors' explanations about statistical analyses are clear enough so that I can validate the code.

Version 1:

Reviewer comments:

Reviewer #1

(Remarks to the Author)

My previous comments have been adequately addressed by the authors, including performing of new analysis on confounding factors that might have impacted the outcome of the study. I therefore recommend acceptance of the revised manuscript.

(Remarks on code availability)

Reviewer #2

(Remarks to the Author)

I appreciate the authors' responses to my queries, and making the effort to incorporate sensitivity analyses adjusting for SES. My only remaining comment is to request the authors to specify how SES was operationalized in the datasets. I don't need to review this again as my major concerns have been addressed.

(Remarks on code availability)

I am not qualified to assess statistical code.

Reviewer #3

(Remarks to the Author)

I am happy that the authors well addressed my previous comments and suggestions. I have no more technical comments but the code seems yet difficult to follow due to redundant parts. Please provide a set of cleaner code files for data cleaning, main analysis, and sensitivity analysis with README files.

(Remarks on code availability)

The provided code is not reader-friendly and seems to have unnecessary parts so that it seems difficult to reproduce the authors' results. It'd be appreciated if data cleaning codes are reduced and the core contents for fitting the hierarchical model are well guided with a README file.

Point-by-point response

Reviewer #1 (Remarks to the Author):

This is a very interesting, hypothesis-generating, case-control study in large patient and control samples from UK health databases, identifying several inflammatory disease entities as potential risk factors for Lung Cancer In Never smokers (LCINS), including discovery and validation cohorts. Although this study is novel, contains original information that are useful for clinicians and may also inform screening strategies for early detection of LCINS, it harbors important caveats that should be adequately addressed before reconsideration:

MAJOR COMMENTS:

1. The Medical Conditions identified to potentially increase the risk of LCINS are quite heterogeneous: Some of them are specific disease entities (ie Lupus, psoriasis), while some others are general medical conditions with multiple aetiologies (ie anemia, GERD) and this renders the results hard to interpret because of the risk of confounding factors. For example, anemia may be idiopathic, but also may be recorded in the context of other systemic disease entities (autoimmune, hematologic, etc). The authors should discriminate between these conditions, acknowledge the limitations of the confounding factors and should stick more to the correlations of specific disease entities, rather than general medical conditions.

We agree with the reviewer, and we mention this as a limitation in the Discussion. Although this is a limitation to our approach, we want to emphasize that our focus is on associations and not causal relationships. This is the largest study investigating medical conditions associated with the occurrence of LCINS, but unfortunately, due to the nature of the data we had to include some heterogeneous diseases. Our analyses are exploratory and more targeted studies are necessary to follow-up on our findings. However, given the reviewers concerns, we attempted to examine heterogeneity of associations for a few conditions within CPRD-Aurum using available codes. Specifically, we identified all anemia and gastritis/colitis related SNOMED codes and assessed associations of LCINS with these disease subtypes. Unfortunately, as shown in Supplemental Table S6, we were underpowered to make firm inference, although iron-related anemia does appear to drive the anemia associations.

Please see below for revised sections in the manuscript:

Methods (page 8):

For select validated conditions that can have multiple etiologies (e.g., anemia), we attempted to use SNOMED codes to identify different manifestations/etiological subtypes of the disease and to then examine associations with LCINS.

Final paragraph in results (page 10):

Of note, iron-related anemia appeared to drive the anemia associations with LCINS (Supplemental Table S6). However, larger studies are needed to confirm these findings.

Discussion (page 13)

Further, some conditions (e.g., anemia and gastritis) can have multiple etiologies and our study was not designed to differentiate between heterogeneous entities.”

...this study was also not designed to examine how individual LCINS risk factors act together to influence LCINS risk and as noted previously several of the conditions co-occur (e.g., autoimmune disease and anemia).

Cancer risk factors rarely act alone and future, more targeted studies can also examine how the relationships between the conditions and medications act in concert to alter LCINS risk.

2. Importantly, the authors unfortunately do not take into account the use of concomitant medications administered for those disease entities, although in the Methods session they clearly state that this information regarding prescriptions was available in the database, at least in the Discovery Cohort (Line 96). This information is extremely important because medications used for many of these conditions (COPD, Lupus, psoriasis, Emphysema, Rosacea) are often immunosuppressive, and this may have contributed to the occurrence of LCINS. It is well known that cancer may arise from an impaired immunity status, especially in the absence of environmental carcinogens, such as smoking. For example, the chronic use of steroids or other immunosuppressive drugs in a non-smoking patient with COPD/asthma or an autoimmune disease can cause severe chronic immunosuppression that can lead to carcinogenesis. In other words, it might be the pharmaceutical immunosuppression that causes LCINS and NOT the autoimmune or inflammatory disease per se. This is an inherent bias of the study that should be adequately addressed and, if authors are not able to adjust for this confounding variable, they should clearly state this important limitation in the Discussion session

We agree with the reviewer that some medications could mediate the associations between medical conditions and LCINS risk. We have obtained data on medication use for both the Aurum and GOLD datasets, linked them to the related conditions and adjusted statistical models for all major findings for use of these medications. The codes for the medications also had to be present within the clinical history of cases identified for the feasibility counts. Some potentially important medications might not be captured within the CPRD (e.g., medications prescribed by a specialist that do not get recorded in the primary care records). Many of the medications were independently associated with LCINS risk, but the additional adjustment did not strongly impact the medical condition-LCINS associations and did not alter any conclusions (see new Table 6).

Please see revised manuscript sections below:

Methods/Sensitivity Analyses (page 7)

We also examined if medications used to treat particular conditions might impact the association estimates of the condition on LCINS risk.¹ We present associations from conditional logistical regression models which include both the condition and the medication(s) that may have been used to treat the condition. UK treatment guidance and the British National Formulary (September 2020-March 2021) were used to compile a list of medications used in treatment of conditions identified as being associated with LCINS Medications were then identified from the records of LCINS cases. We required only a single medication prescription within the 1-10 years for an individual to be considered exposed to the medication (ever use).

Results/Sensitivity Analyses (page 10).

We adjusted for medication use (ever use) in both the discovery and validation analyses for COPD, GERD and gastritis/NIIGC. We only adjusted for medication use for psoriasis, lupus and rosacea in CPRD-GOLD, because they were not statistically associated with LCINS in CPRD-Aurum. We examined use of the following medications with codes available in both, CPRD-GOLD and CPRD-Aurum: oral corticosteroids; inhaled corticosteroids; methotrexate; other immunosuppressants; Long-Acting Beta-2 Agonists (LABA); Short-Acting Beta-2 Agonists (SABA); Antimuscarinic Bronchodilators; macrolides; proton pump inhibitors (PPI); H2-receptor antagonists; and antacids; non-steroidal anti-inflammatory drugs; and tetracyclines (Table 6). All codes used to classify medications are in supplemental material DiscoveryMedications.xlsx and ValidationMedications.xlsx. Associations between LCINS and the conditions were only modestly attenuated when medications were added to the statistical models.

Discussion: (page 11).

Adjusting for several of the medications potentially used in the treatment of COPD modestly attenuated the association, suggesting that they might mediate some of the association with COPD.

Discussion: (page 12).

PPIs, H2 receptor antagonists and antacids were independently associated with LCINS risk. PPI use has been linked to various cancers, including gastric cancer in the CPRD.⁵⁴ Several mechanisms by which PPIs might contribute to cancer development have been proposed, including alteration of the gut microbiome which can affect inflammation, immune responses, and the production of carcinogenic metabolites by overgrowth of harmful bacteria. PPI-induced hypergastrinemia can cause proliferation and hyperfunction of enterochromaffin-like (ECL) cells.⁵⁵ Our observations could also be driven by vague “symptoms” that are due to a latent, yet undetected LCINS, which led to antacid or PPI/H2 receptor antagonist use.

Discussion: (page 13).

Severe psoriasis is treated with methotrexate which can cause pulmonary fibrosis.^{57,58} The association between psoriasis and LCINS was modestly attenuated when methotrexate use was added to the statistical model.

The elevated risk for lupus in CPRD-GOLD that was not replicated in CPRD-Aurum may have been driven by systemic lupus erythematosus (SLE) which can affect the lungs and is treated by immunosuppressant medications such as azathioprine (Table 6).

3. The title of the Manuscript can be misleading, leading to a misinterpretation that all kinds of inflammations are associated with LCINS. I would suggest to modify the title as “Inflammatory diseases and risk of lung cancer among never smokers”

We have revised the manuscript title as suggested by the reviewer to **“Inflammatory diseases and risk of lung cancer among individuals who have never smoked”**.

4. Finally, this manuscript contains high-magnitude statistical analysis and this Reviewer is not competent to perform accurate review of the Statistical Methods session. I would therefore advise Official Statistical Review of the study by a statistician

MINOR COMMENTS:

1. Line 107: What was the rate of discrepancy between the smoking codes?

We do not know the exact discrepancy between the smoking codes, but smoking prevalence in CPRD aligns with UK smoking estimates generally.^{2,3} To maximize the chances of correctly identifying smoking status we used the entire clinical history and excluded individuals without documented smoking status or those with a conflicting history. Additionally, the prevalence of COPD (1%) in the controls (who were selected using identical criteria as for the cases) is what would be expected given the proportion of COPD occurring in never smokers in the UK (20-25%) and the expected prevalence of COPD in persons >35 years of age in the CPRD.

2. Line 112: What about other HPV-related cancers? (Oropharyngeal, anal, etc)

The reviewer makes a good point. We opted to include only HPV-related cervical cancer cases because cervical cancer is more common than the others, very rare, HPV-related cancers. However, there were only n=2 cervical cancer cases and we did not further analyze them, as we required each disease category to include at least 15 cases (see response to reviewer 3 below).

3. Line 132: Why was obesity excluded from the analysis as a potential risk factor for LCINS? Please provide justification as you did for alcohol consumption (exposure to passive smoking)

Unfortunately, obesity is not uniformly recorded across practices. However, following the reviewer’s request, we added new results from models that additionally adjusted for BMI (where available) as a potential confounding factor for the association between inflammatory diseases and LCINS risk, and the results were

not substantially changed (see results in new Table 5 where we jointly adjusted for both BMI and SES in Aurum and Supplemental Table 5 where models are jointly adjusted for BMI and SES in CPRD-GOLD).

4. Line 267: How do authors explain that there were no cases of recorded IPF in thousands of cases and controls studied? Is it possible that this disease code was not captured as a disease entity in the databases used?

We were also initially surprised by not observing any IPF from the feasibility counts. However, after further research we learned that IPF is very rare in the UK with a prevalence = 0.78 per 10,000 persons (0.38, 1.,63). We had 1478 individuals in our feasibility counts and thus did not expect to observe even one IPF case among them. We comment on that point in the revised Discussion.

Reviewer #2 (Remarks to the Author):

This report presents the results from a comprehensive, though exploratory, analysis of EHR data compiled from two substantial databases, to examine associations of various comorbidities with lung cancer in never smokers (LCINS). The discovery dataset covers 41 primary care practices within the United Kingdom's Clinical Practice Research Datalink, called CPRD-GOLD, comprising >20 million residents, and the validation dataset is a similar but independent dataset called CPRD-Aurum, covering selected regions within the UK and 40 million residents. The authors examined a large number of conditions, some considered to have established or putative associations with LCINS. This is a methodologically rigorous analysis, with multiple sensitivity analyses and statistical approaches taken to mitigate possible biases and false discoveries. The results confirm previously established associations, and provides additional data points for putative risk factors. The newly identified risk factors are considered hypothesis generating findings. I had just two questions that the author might address.

1. The study design was a nested case-control analysis, and was robustly attentive to most possible sources of bias inherent to the data. The authors may want to comment on why they did not conduct a longitudinal time-to-event analysis using all records.

A full cohort analysis with careful adjustment for potential confounders would certainly have been a valid alternative to the case-control study design. However, given the large size of the dataset, the latency considerations in the exposures that require time varying covariate coding and analyses, and the many conditions we evaluated, the computational burden of the prospective approach would have been substantial. Careful matching of cases and controls and the retrospective design allowed us to control for confounders already in the design stage, and the retrospective design made it easier to accommodate the latency period for the conditions, lessening the computational effort.

2. As the authors themselves explain, LCINS has a strong association with socioeconomic status. To the extent that some of the conditions are also associated with socioeconomic status, the authors should comment on whether it is possible that some of the associations may be due to unmeasured confounding by SES. Is it possible to adjust for some indicator of SES in these analyses?

We did not originally adjust for SES in the CPRD-GOLD database because we anticipated more than 50% of the participants would lack linkage to SES information. After obtaining linkages to SES information, we re-ran the analysis among those study participants with SES information (690 never smoking lung cancer cases and 6,285 never smoking controls had SES data, comprising 43% in GOLD). Jointly adjusting for both SES and BMI did not materially alter our associations (Supplemental Table S5) although the sample size was greatly diminished. In CPRD-Aurum, SES data were available for the entire population through linkage to the index of multiple deprivation complete (Table 5). Associations from conditional logistic models jointly adjusted for both BMI and SES in Aurum remained unchanged. Please see revised Methods (Page 7, 8) and Results (Page 10) sections.

Reviewer #3 (Remarks to the Author):

The manuscript entitled “Inflammatory diseases are associated with lung cancer in never smokers” is a hypothesis-generating study trying to identify risk factors for LCINS. Remarkably, the authors presented other recent studies supporting their results, suggesting that inflammatory diseases including COPD or GERD would be parts of LC pathogenesis. This work could broaden a perspective to establish screening programs for LC early detection. It is well-written overall, however, I have a few concerns mainly on the statistical analyses.

Line 121 explained the study conducted in 2018 however the analysis was done additionally with 2019 data as stated in Line 182. Unless there are reasons why it has to be, it'd be good to introduce exposure classification

Number of Conditions to examine	Prevalence in cases	Alpha (0.05/Number of conditions)	Power	Minimum Detectable OR
50	1%	0.001	80%	2.30
50	5%	0.001	80%	1.59
50	10%	0.001	80%	1.40
50	20%	0.001	80%	1.30
100	1%	0.0005	80%	2.14
100	5%	0.0005	80%	1.49
100	10%	0.0005	80%	1.35
100	20%	0.0005	80%	1.27

with the full data explored.

We apologize for the confusion. We obtained approval for the study in late 2018 after obtaining preliminary feasibility counts and performing power calculations. By the time we were close to developing full Read code lists and grouping conditions into clinically meaningful categories it was almost 2019 so we requested an additional year of data. We have clarified this point in the revised manuscript.

Please elaborate more on the arguments about ‘statistical power’ (Lines 125, 129). What was the authors’ statistical power? How do the case prevalence and the categorizing strategy relate to the power?

Prior to submitting our proposal to CPRD to perform the study, we examined the statistical power under a range of assumptions including potential effect size, numbers of conditions we could evaluate assuming a multiple testing (Bonferroni) corrected alpha level. We generated the table with detectable odds ratios given here. As shown in the table, detecting associations for diseases present in <1% cases was not reasonable given only~1500 expected cases. This table was submitted with our proposal to CPRD, and we now include it as a supplemental table in the manuscript. We have further clarified the text in the exposure classification section as below.

Methods/ Exposure classification for the discovery stage (Page 5).

We required each disease category (called “subcategory” below) to include at least 15 cases, corresponding to a case prevalence of ~1% ($0.01 \times 1,478 \text{ LCINS} = \sim 15$). This was based on statistical power calculations under a range of reasonable assumptions, including a multiple testing corrected alpha level (Supplemental Table S1).

From our original protocol available on CPRD website:

Study power will be dependent on both the number and prevalence of conditions and medications within the population. Based on our preliminary estimates, we will have 80% power to detect odds ratios from approximately 1.27 to 2.30, accounting for multiple comparisons, depending on the prevalence of each factor in the case group.

Line 132 stated that obesity or alcoholism conditions were not considered. I understand that alcoholism-related conditions were excluded because alcohol consumption might be associated with passive smoking as the authors explained. However, what is the reasoning to exclude obesity-related conditions?

See prior response to Reviewer 1 on this issue.

The authors called the study using individuals in the 1-10 year period prior to selection as a primary study because it has a large sample and is more representative of the UK population. It is no wonder that the larger sample size is favorable for the higher power, however, how come it is more representative?

Using individuals with more than 10 years of primary care data in the analysis restricts the study population to older subjects who may have more comorbidities and different behaviors, not representative of the UK general population.

The manuscript would be enhanced if the two-stage structure of the primary (conditional logistic) and secondary (hierarchical; logistic) models were more clearly stated. Currently, terminologies are mixed so it wasn't easy to understand how the models were fitted. What is "this strategy" in Line 128 that enables the fitting of statistical hierarchical models? Does the "two-phase" approach indicate the primary-secondary structure based on broader/sub- categories, or 1-10 year plus 10-32 year analysis to explore whether the associations identified in 1-10 years persisted more than 10 years? What is the motivation of including random effects in which subcategories vary randomly around the mean of the primary category? What does it mean that the ORs of second stage model are "mutually adjusted", and how come does it lead to $p < 0.05$ being considered as significant? Why the matching factors are adjusted in the hierarchical model even though the data are from a nested case-control study? Why subcategories (peripheral vascular and myocardial infarction) were fitted even though its primary category (cardiovascular) was not strong enough, which contradicts the explanations in Lines 153-155? More elaboration on mathematical representations in the appendix and/or relevant references would help for this purpose.

We have revised the description of the categorization of the conditions to be clearer about the hierarchical structure. We use the expressions "category" and "primary disease category" for the larger groups and "subcategory" for the individual conditions within a larger group. We use the term "two-stage statistical approach" throughout the manuscript. We have also revised the technical appendix that describes the hierarchical modeling to improve clarity. We have added greater detail as to why certain analyses were performed.

For each primary disease category that was associated with LCINS at a false discovery rate (FDR) < 0.05 , we fit the hierarchical logistic regression model that included all individual conditions in that primary category. This approach greatly reduces the number of tests that are performed, as in the first stage, only 24 comparisons are considered. The hierarchical model jointly estimates the associations of all subcategories within a larger primary disease category and thus no multiple testing adjustment is required. However, as there is no software that implements a conditional logistic model for hierarchical regression, we used unconditional logistic regression and adjusted for the matching factors to accommodate the matched design. This approach yields valid inference (see e.g. Chia-Ling K et al, *Frontiers in Public Health*, 2018; URL=<https://www.frontiersin.org/journals/public-health/articles/10.3389/fpubh.2018.00057>)

As a sensitivity analysis we also present association estimates for all individual subcategories from conditional logistic regression at both time points. We also did this so that the reader could contextualize all associations within broader scientific knowledge. For example, asthma (within the allergies category) was not significantly associated with LCINS in our study, despite commonly being associated with LCINS in the literature.

In Statistical Analyses:

"We used a two-step approach to identify medical conditions associated with LCINS in the 1-10 years before diagnosis or selection. First, we estimated adjusted odds ratios (aORs) and 95% confidence intervals (CIs) for all 24 primary disease categories (coded as 1 if any condition in that category was present in the clinical history 1-10 years prior to diagnosis/selection and 0 otherwise) using conditional logistic regression models to account for the matched design. As age was used in categories for the matching. we also adjusted for age at selection in single years. For each primary disease category that was associated with LCINS at a false discovery rate (FDR) < 0.05 , we then fit a hierarchical (random effects) logistic regression model (second step) that included

all individual conditions included in that primary category. Under that model the odds ratios of the individual conditions that comprise the primary category are assumed to vary randomly around the log-OR of the primary category (details are given in Supplemental document S1Hierarchical.doc). We adjusted the hierarchical models for matching factors to accommodate the matched study design. Because the ORs of the individual conditions are mutually adjusted for all other conditions in the primary category under the hierarchical model, $p < 0.05$ was considered statistically significant.”

I wonder for what purpose the sensitivity analyses were performed and how. (Line 160)

We used conditional logistic regression to fully accommodate the matched design for the individual conditions as a sensitivity analysis for the hierarchical models. Conditioning on the matching variables also removes the effect of any unmeasured confounders correlated with the matching variables and thus is a more robust, although somewhat less efficient statistical analytic approach. We conducted secondary analyses to test associations of LCINS with longer lag periods. We have updated the text in the methods because our rationale was unclear. We also conducted sensitivity analyses for LCINS associations adjusting for SES, BMI and selected medications. We created a separate section in the Statistical analyses describing these approaches.

Sensitivity analyses (Pages 6-7)

To assess the robustness of the hierarchical regression model results, we examined associations between all primary categories and individual conditions in both time intervals and LCINS risk using conditional logistic regression. Conditioning on the matching variables removes the effect of any unmeasured confounders correlated with the matching variables and thus is a more robust, although less efficient statistical analytic approach. The aORs from conditional logistic models are not adjusted for other individual conditions in the same disease category. We used conditional logistic regression to examine the 1-10 year condition-LCINS associations additionally adjusted for body mass index (BMI, coded as underweight, normal, overweight, obese, missing) and socioeconomic status (SES) for those conditions which were statistically significant in the discovery stage. SES is associated with many diseases and has been shown to be independently associated with LC risk.²² SES in deciles was available for individuals who were linkable to the CPRD index of multiple deprivation.²³ In CPRD-GOLD, 43% of the study population had linkage to SES information. BMI, which has been more consistently recorded over time²⁴, was available for ~25% of the study population. The most recent BMI measurements were used. Measurements occurring > 15 years before index date were set to missing.

We also examined if medications used to treat particular conditions might impact the association estimates of the condition on LCINS risk.²⁵ We present associations from conditional logistical regression models which include both the condition and the medication(s) that may have been used to treat the condition. UK treatment guidance and the British National Formulary (September 2020-March 2021) were used to compile a list of medications used in treatment of conditions identified as being associated with LCINS. Medications were then identified from the records of LCINS cases. We required only a single medication prescription within the 1-10 years for an individual to be considered exposed to the medication (ever use).”

The authors said the validation set does not capture some regions so that it is not geographically representative (Line 169). Can this explain any of the different result from the discovery study, such as the opposite direction of Lupus in Fig 1?

Besides having very small numbers in both analyses because Lupus is rare, we do suspect that geographic heterogeneity could explain this difference. We have expanded upon this in the **Discussion (page 13)**:

The elevated risk for lupus in CPRD-GOLD that was not replicated in CPRD-Aurum may have been driven by systemic lupus erythematosus (SLE) which can affect the lungs and is treated by immunosuppressant medications such as azathioprine (Table 6). SLE is more common in Northern Ireland and Scotland, two regions not represented in the validation dataset.

It is unclear in Lines 289-291 that 1% COPD in the controls supports the good quality of smoking assessment.

We feel confident in the quality of smoking assessment, because the proportion of COPD observed in the never-smoking controls (1%) is approximately what is expected given that ~20-25% of COPD cases in the UK occur in never-smokers, and the prevalence of COPD in all individuals >35 years of age is estimated ~4%-5% given COPD codes documented in the CPRD.^{4,5} However, to avoid confusions, we opted to delete this sentence from the manuscript.

The Tables 2 and 4 are not fully shown. Please update the right columns that show p-values and/or FDR.

Thank you for identifying the formatting problem. We have fixed this mistake.

In the tables, NR stands for “not reported” due to small cell size (<5) however there are cells stating “<5” instead of NR. Please clean up the notations.

Thank you for identifying the inconsistencies. We have fixed them.

Please clarify how the validation was performed so that it is persuasive. Was the primary model for the validation set fitted but not reported, or not fitted but the secondary model can be fitted based on the discovery set analysis? Which stage is for “...used conditional logistic regression... in the validation phase” in Line 172 (if the second, is it the right way?)?

We have clarified the validation strategy throughout the manuscript and hope it is now clear to the reader. We only validated individual (not category) associations, using conditional logistic regression. We did not consider a multiple testing correction in the validation, but instead present all information and focus on qualitative similarities between associations from the discovery and validation data.

Methods (Page 4)

We used a two-stage approach. In the first stage we identified conditions associated with LCINS risk using CPRD-GOLD. Then, we validated these conditions in an independent dataset, CPRD-Aurum.

Independent validation in CPRD-Aurum (Pages 7-8)

Specific conditions (not primary categories) that were significantly associated ($p < 0.05$) with LCINS in the primary analysis (1-10 years) in CPRD-GOLD were validated in the independent CPRD-Aurum dataset. CPRD-Aurum is a population-based research quality database covering 10 regions of England.¹⁸ Case and control selection was identical to the discovery stage. Individuals who were present in both databases (23%) were excluded from the validation dataset. We also removed controls who lost matched cases in the deduplication process. Unlike CPRD-GOLD, CPRD-Aurum does not capture patients in Northern Ireland, Scotland or Wales and is therefore not geographically representative of the entire UK population.²⁶ For each condition we validated, we identified the earliest date at which the condition was observed in the 1-10 and 10-32 years prior to the index date in CPRD-Aurum. Conditions only identified within the year before the index date were excluded to reduce the potential for reverse causation. SNOMED codes were used to classify conditions in CPRD-Aurum. Conditional logistic regression models were used to estimate aORs between conditions and LCINS both the 1-10 and 10-32-year assessment windows. We present all associations examined the validation data but focus on those which were significant ($p < 0.05$) in CPRD-Aurum. As in the discovery stage we also performed sensitivity analyses in which conditional regression models were jointly adjusted for BMI and SES for significant findings. In contrast to the CPRD-GOLD population, most of the CPRD-Aurum population is linkable to SES information, thereby providing a more robust interpretation of the results. Following a similar strategy in the discovery stage, we also adjusted for medication use in statistical models for conditions associated with LCINS. See supplemental material ValidationMedications.xlsx for medications. For select validated conditions that can have multiple etiologies (e.g., anemia), we attempted to use SNOMED codes to identify different manifestations/etiological subtypes of the disease and to then examine associations with LCINS.

References

- 1 Tchetgen Tchetgen, E. J. & Phiri, K. Evaluation of Medication-mediated Effects in Pharmacoepidemiology. *Epidemiology* **28**, 439-445 (2017). <https://doi.org/10.1097/EDE.0000000000000610>
- 2 Herrett, E. *et al.* Data Resource Profile: Clinical Practice Research Datalink (CPRD). *Int J Epidemiol* **44**, 827-836 (2015). <https://doi.org/10.1093/ije/dyv098>
- 3 Booth, H. P., Prevost, A. T. & Gulliford, M. C. Validity of smoking prevalence estimates from primary care electronic health records compared with national population survey data for England, 2007 to 2011. *Pharmacoepidemiol Drug Saf* **22**, 1357-1361 (2013). <https://doi.org/10.1002/pds.3537>
- 4 Lamprecht, B. *et al.* COPD in never smokers: results from the population-based burden of obstructive lung disease study. *Chest* **139**, 752-763 (2011). <https://doi.org/10.1378/chest.10-1253>
- 5 Rayner, L., Sherlock, J., Creagh-Brown, B., Williams, J. & deLusignan, S. The prevalence of COPD in England: An ontological approach to case detection in primary care. *Respir Med* **132**, 217-225 (2017). <https://doi.org/10.1016/j.rmed.2017.10.024>

Point-by-point responses to reviewers' comments

Reviewer #1 (Remarks to the Author):

My previous comments have been adequately addressed by the authors, including performing of new analysis on confounding factors that might have impacted the outcome of the study. I therefore recommend acceptance of the revised manuscript.

Response: We thank the reviewer for their time and constructive feedback.

Reviewer #2 (Remarks to the Author):

I appreciate the authors' responses to my queries, and making the effort to incorporate sensitivity analyses adjusting for SES. My only remaining comment is to request the authors to specify how SES was operationalized in the datasets. I don't need to review this again as my major concerns have been addressed.

Reviewer #2 (Remarks on code availability):

I am not qualified to assess statistical code.

Response: We thank the reviewer for their time and constructive feedback. SES was an integer that mapped to deciles of social deprivation (index of multiple deprivation). It was included in the models as a linear term. We have now clarified this point in the Methods section.

Reviewer #3 (Remarks to the Author):

I am happy that the authors well addressed my previous comments and suggestions. I have no more technical comments but the code seems yet difficult to follow due to redundant parts. Please provide a set of cleaner code files for data cleaning, main analysis, and sensitivity analysis with README files.

Reviewer #3 (Remarks on code availability):

The provided code is not reader-friendly and seems to have unnecessary parts so that it seems difficult to reproduce the authors' results. It'd be appreciated if data cleaning codes are reduced and the core contents for fitting the hierarchical model are well guided with a README file.

Response: We have now updated our codes and the README file in this revised submission to make them more friendly to readers. We thank the reviewer for their time and constructive feedback